# Euler–Poincaré Neural Dynamics:
# A Geometric-Mechanics Framework for Scientific Simulation

**Sungwoo Park** [1] [†]   **Jongwon Lee** [1]   **Jiwoong Kim** [1]

## Abstract

We introduce Euler–Poincaré Neural Dynamics (EPND), a geometric-mechanics framework that casts evolution-operator learning as Lie-group flows for long-horizon dynamical modeling. Unlike conventional operator-learning approaches that treat temporal propagation as an unconstrained black-box map, EPND places geometric mechanics at the core of its architecture, playing a role of the mathematical engine. This foundation enables a principled treatment of curvature, symmetry, and conservation, with the learned evolution expressed in geometric terms. Building on this foundation, we develop the Euler–Poincaré Parallel Scan, a parallel algorithm that leverages the associative algebra of Lie-group compositions to overcome the inefficiencies of sequential computation. By unifying geometric structure with scalable computation, EPND achieves high accuracy, strong stability, and significant parallel acceleration in modeling long-horizon dynamics in versatile scientific simulations.

## 1. Introduction

Geometric mechanics possesses a deep and enduring intellectual lineage, tracing back to Euler's foundational formulations of rigid-body motion and stability, and later enriched by Poincaré's geometric perspective on celestial mechanics. Its modern mathematical form was rigorously developed by Arnold and Marsden Arnold (1989); Marsden & Ratiu (1999), who recast classical mechanics in the differential-geometric language of manifolds, Lie groups, and connections. By elucidating the intrinsic geometric structures underlying physical phenomena, geometric mechanics provides a unified framework linking curvature, symmetry, and conserved quantities in governing system behavior. These principles have become fundamental across diverse scientific fields, including high-dimensional molecular simulations Janezic & Merzel (1995); Caillol (2020); Markiewicz (1999), plasma physics Scovel & Weinstein (1994); Zhang et al. (2016); Kraus (2013), and astrophysics and general relativity Bardeen (1970); Berti et al. (2015); Tsang et al. (2015), demonstrating the maturation of geometric mechanics into a broadly unifying paradigm in modern science.

Motivated by these successes, recent efforts within the ML community have also sought rigorous interpretations for analyzing complex nonlinear dynamical systems. One prominent example is Koopman operator theory Koopman (1931); Mezić (2005); Rowley et al. (2009); Schmid (2010); Kutz et al. (2016); Brunton & Kutz (2022), a principled, operator-theoretic approach widely adopted to represent nonlinear evolution through linear operators acting on observables. However, despite substantial progress in developing Koopman-based methods, a comprehensive treatment explicitly rooted in geometric mechanics remains notably absent. This gap highlights a valuable opportunity: systematically leveraging geometric principles to enhance data-driven dynamics system modeling frameworks.

Following this historical and conceptual trajectory, we introduce *Euler–Poincaré Neural Dynamics* (EPND), which builds on a Koopman-style perspective to model state evolution within the geometric-mechanics paradigm. While geometric mechanics provides an elegant formulation of nonlinear dynamics through Lie groups, the associated computations of Koopman-type operators become prohibitively complex and inefficient when performed sequentially, leading to significant computational bottlenecks for long-horizon tasks. To overcome this fundamental inefficiency, EPND introduces a new parallel algorithmic structure termed the *Euler–Poincaré Semidirect Scan*, which leverages the algebraic properties of Lie-group flows to drastically enhance computational scalability and efficiency. By interpreting Koopman-type evolution as geometric flows induced by intrinsic velocity fields on Lie groups, this framework not only yields a geometric representation of the evolution, but also achieves substantial parallel speedups over conventional sequential methods. Motivated by these considerations, we

---

[†]Corresponding author.   [1]Department of Computer Science and Engineering, Korea University. Correspondence to: Sungwoo Park <sungwoo_park@korea.ac.kr>.

*Proceedings of the 43rd International Conference on Machine Learning*, Seoul, South Korea. PMLR 306, 2026. Copyright 2026 by the author(s).

specifically highlight two main contributions:

1. We propose Euler–Poincaré Neural Dynamics (EPND), a geometric-mechanics parameterization of Koopman-style evolution operators on Lie groups. EPND embeds evolution operators within intrinsic velocity fields on Lie groups, creating a structured grey-box model that inherently respects algebraic symmetries through geometric transport.

2. Recognizing computational bottlenecks associated with sequential Lie-group computations, we introduce a novel parallel computational strategy that exploits the associativity of geometric operations to significantly improve computational efficiency and enable scalable long-horizon simulation of complex dynamical systems.

## 2. Method: Euler–Poincaré Neural Dynamics

Our framework centers on recasting dynamical evolution in the language of geometric mechanics. To set the stage, we first revisit the conventional operator-learning formulation for dynamical systems as follows.

**Motivation: Operator Learning.** Let $\mathcal{X} \subset \mathbb{R}^d$ be the state space, and consider an absolutely continuous trajectory $\mathbf{x}(\cdot)$ evolving under a given vector field $\mathbf{v}(\cdot, \cdot)$:

$$\dot{\mathbf{x}}(t) = \mathbf{v}(t, \mathbf{x}(t)), \quad \mathbf{x}(0) = \mathbf{x}_0 \in \mathcal{X}. \quad (1)$$

We define the flow map $\Phi_{s,t} : \mathcal{X} \to \mathcal{X}$ that transports the state from time $s$ to time $t$, satisfying $x(t) = \Phi_{s,t}(x(s))$ and the two-parameter semigroup property $\Phi_{s,t} \circ \Phi_{r,s} = \Phi_{r,t}$ for all $r \leq s \leq t$. An operator-theoretic viewpoint represents the nonlinear evolution in (1) linearly in an infinite-dimensional space of observables. Specifically, the *evolution operator* $\mathcal{U}_{s,t}$ is defined by its action on observables $f : \mathcal{X} \to \mathbb{R}$ via composition with the flow map for $s \leq t$, $(\mathcal{U}_{s,t}f)(x) := f(\Phi_{s,t}(x))$. Consequently, the family of evolution operators $\{\mathcal{U}_{s,t}\}_{s \leq t}$ constitutes a linear two-parameter semigroup, satisfying:

$$\mathcal{U}_{s,t}\mathcal{U}_{r,s} = \mathcal{U}_{r,t}, \quad \mathcal{U}_{t,t} = \text{Id}.$$

This viewpoint allows the transformation of nonlinear state dynamics into linear operator dynamics on function spaces such as $C^1(\mathcal{X})$. For $f \in C^1(\mathcal{X})$ define the time–dependent generator by $(\mathcal{L}_t f)(x) := \nabla f(x) \cdot v(t, x)$. By differentiating along $t$, one obtains the *operator evolution equation*

$$\frac{\partial}{\partial t}(\mathcal{U}_{s,t}f) = \mathcal{U}_{s,t}(\mathcal{L}_t f), \qquad \mathcal{U}_{s,s} = \text{Id}. \quad (2)$$

From this operator perspective, the evolution of an initial state $\mathbf{x}_0 \in \mathcal{X}$ at time 0 to its subsequent state $\mathbf{x}_t$ at time $t$ can be expressed implicitly through its action on coordinate functions. Specifically, let the identity observable be

denoted by $\iota : \mathcal{X} \to \mathcal{X}$, $\iota(x) = x$. Then, the state evolution from $\mathbf{x}_0$ to $\mathbf{x}_t$ can be expressed explicitly as:

$$\mathcal{U}_{0,t}[\iota](\mathbf{x}_0) = \big(\Phi_{0,t}^1(\mathbf{x}_0), \ldots, \Phi_{0,t}^d(\mathbf{x}_0)\big) = \Phi_{0,t}(\mathbf{x}_0) = \mathbf{x}_t.$$

This illustrates how the nonlinear state evolution is recovered through the action of the evolution operator on the identity observable $\iota$. By parameterizing the evolution operator $\mathcal{U}_{0,t}^\theta$ with neural networks $\theta \in \Theta$, previous works represent the dynamics via a coupled evolution system involving both the state trajectory and its associated time-derivative:

$$\dot{\mathbf{x}}(t) = \big(\mathcal{L}_t \iota\big)(\mathbf{x}_t), \quad \mathbf{x}_t = \big(\mathcal{U}_{0,t}^\theta[\iota]\big)(\mathbf{x}_0). \quad (3)$$

The primary object of this work is then to learn the best parameterization $\mathcal{U}^\theta$ that captures accurate dynamics. In other words, given an initial datum $\hat{\mathbf{x}}_0$, our goal is to accurately predict its future deterministic trajectory:

$$\mathcal{J}(\theta) := \mathbb{E}_{\hat{\mathbf{x}} \sim \mathbb{P}_{\text{data}}} \left[ \int_{[0,T]} \big\| \big(\mathcal{U}_{0,t}^\theta[\iota]\big)(\hat{\mathbf{x}}_0) - \hat{\mathbf{x}}_t \big\|^2 dt \right], \quad (4)$$

While this objective provides a direct learning signal, long-horizon rollouts remain difficult when the evolution operator is modeled as an unconstrained black-box map, leading to compounding errors and loss of physical fidelity. We therefore replace black-box parameterizations with a geometric-mechanics one, realizing the learned evolution operator as a Lie-group action on observables and turning operator learning into a structured grey-box model with explicit algebraic and physical constraints. Our aim is to implement and analyze long-horizon evolution through these geometric-mechanics constructs, rather than to derive new operator-theoretic results.

**Geometrification of Koopman Dynamics.** Our primary aim is then to reformulate the conventional perspective on Koopman dynamics into a rigorous geometric framework, enabling a deeper analysis and discussion through concepts drawn from geometric mechanics. Let $G$ be a Lie group with identity element $e$ and $\mathfrak{g} := T_e G$ is the corresponding Lie algebra equipped with the bracket $[\cdot, \cdot]$. For any element $g \in G$, the adjoint action $\text{Ad}_g : \mathfrak{g} \to \mathfrak{g}$ is defined as the differential at the identity of the conjugation map. Its infinitesimal counterpart, the adjoint representation $\text{ad}_\xi : \mathfrak{g} \to \mathfrak{g}$, is explicitly given by $\text{ad}_\xi(\eta) = [\xi, \eta]$ for any $\eta \in \mathfrak{g}$. We endow $\mathfrak{g}$ with an inner product $\langle \cdot, \cdot \rangle_\mathfrak{g}$ and fix a symmetric positive–definite *inertia operator* $\mathbb{I} : \mathfrak{g} \to \mathfrak{g}^*$. Given any curve $g(t) \in G$ on a Lie group and the left translation map $L_g : G \to G$ defined by $L_g(h) = gh$, we define the left–trivialized velocity as

$$\xi(t) := [L_{g(t)^{-1}}]_* \dot{g}(t) = g(t)^{-1}\dot{g}(t) \in \mathfrak{g}.$$

To study the variations of curves on the Lie group $G$, we introduce a *variation* $\delta g(t) = \frac{d}{d\epsilon}\big|_{\epsilon=0} g_\epsilon(t)$, where $g_\epsilon(t)$ is

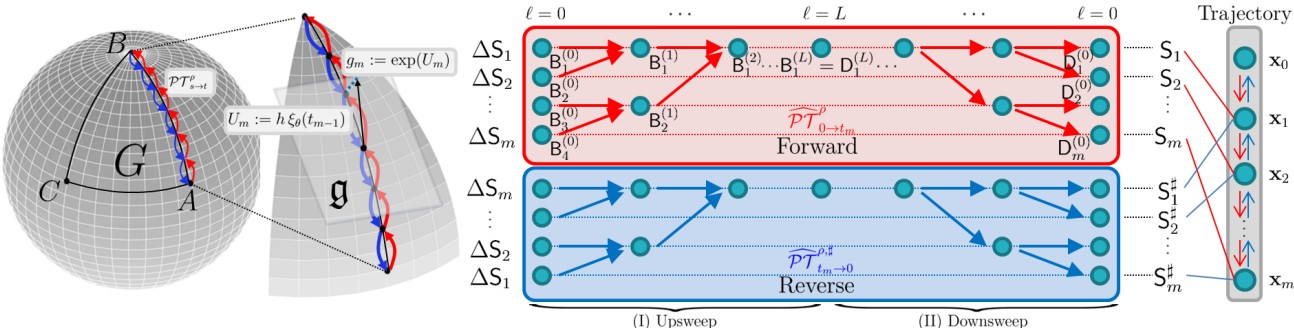

**Figure 1. Overall Pipeline.** *(left)* A neural geodesic field defines infinitesimal motions on the Lie algebra, which exponentiate into discrete steps $g_m$ tracing a geodesic on the group manifold. *(right)* These steps yield $\Delta S_m$, which are computed by a prefix scan algorithm with upsweep and downsweep phases, resulting in the cumulative states $S_m$ or $S_m^\sharp$ as in Appendix C. The resulting states propagate trajectories in a reversible and geometrically consistent manner, providing scalable forward/reverse evolution across long time horizons.

a smooth one-parameter family of curves satisfying $g_0(t) = g(t)$ with fixed endpoints. This induces the corresponding variation:

$$\eta(t) := g(t)^{-1}\delta g(t) \in \mathfrak{g}, \quad \eta(0) = \eta(T) = 0.$$

Under these definitions, the variation of the velocity vector $\xi(t)$ in the Lie algebra is captured succinctly by the *Euler–Poincaré variation formula* Holm et al. (1998): $\delta\xi(t) = \dot{\eta}(t) + [\xi(t), \eta(t)]$. Next, we consider a left–invariant Lagrangian $L : TG \to \mathbb{R}$ which, due to left invariance, reduces naturally to a function $\ell : \mathfrak{g} \to \mathbb{R}$ on the Lie algebra. In particular, by selecting a quadratic kinetic energy-type Lagrangian, we specify

$$\ell(\xi) = \frac{1}{2}\langle \xi, \mathbb{I}(\xi)\rangle_\mathfrak{g}, \quad m(t) := \frac{\delta\ell}{\delta\xi}(t) = \mathbb{I}(\xi) \in \mathfrak{g}^*, \quad (5)$$

where $\mathbb{I} : \mathfrak{g} \to \mathfrak{g}^*$ is an inertia operator defined via the inner product. Next, applying Hamilton's principle to the action functional associated with this left–invariant Lagrangian, $\mathcal{S}[g] = \int_0^T \ell(g^{-1}\dot{g})dt$ and imposing variations with fixed endpoints, one arrives at the *Euler–Arnold equation* Holm et al. (1998); Modin et al. (2010) on Lie algebra $\mathfrak{g}$:

$$\frac{d}{dt}\frac{\delta\ell}{\delta\xi}(t) + \mathrm{ad}_\xi^*\frac{\delta\ell}{\delta\xi}(t) = \dot{m}(t) + \mathrm{ad}_\xi^* m(t) = 0. \quad (6)$$

In this Lagrangian-derived expression, $\mathrm{ad}_\xi^* : \mathfrak{g}^* \to \mathfrak{g}^*$ denotes the coadjoint action, which is defined through the adjoint equation $\langle \mathrm{ad}_\xi^* m, \eta \rangle_\mathfrak{g} = \langle m, [\xi, \eta] \rangle_\mathfrak{g}$ for all $\eta \in \mathfrak{g}$. The natural geometric structure of a Lie group endowed with a left-invariant metric induced by an inner product on its Lie algebra gives rise to the connection on left-invariant fields as shown in Proposition B.1:

$$\nabla_X^\mathfrak{g} Y = \frac{1}{2}\left([X, Y] - \mathrm{ad}_X^\dagger Y - \mathrm{ad}_Y^\dagger X\right),$$

where the adjoint operator $\mathrm{ad}_\xi^\dagger$ is explicitly defined by

$$\mathrm{ad}_\xi^\dagger = [\mathbb{I}^{-1}] \circ \mathrm{ad}_\xi^* \circ [\mathbb{I}], \quad \langle \mathrm{ad}_\xi^\dagger u, v \rangle_\mathfrak{g} = \langle u, \mathrm{ad}_\xi v \rangle_\mathfrak{g}.$$

Under this formulation, the Euler-Arnold equation in (6) on the Lie algebra $\mathfrak{g}$ reduces precisely to the geodesic equation associated with the Levi–Civita connection $\nabla^\mathfrak{g}$:

$$\dot{\xi}(t) = -\mathrm{ad}_\xi^\dagger \xi, \quad \xi = g^{-1}(t)\dot{g}(t), \quad \nabla_{\dot{g}}^\mathfrak{g}\dot{g} = 0. \quad (7)$$

When the Lie group acts on an external representation space (*i.e.*, vector space $V \simeq \mathbb{R}^d$) via a group representation $\rho : G \to \mathbb{M}(V)$ with its differential $\rho_* : \mathfrak{g} \to \mathfrak{m}(V)$, this notion of transport naturally extends to $V$. The induced operator called a *parallel transport*, denoted $\mathcal{PT}_{s\to t}^\rho$, can be obtained by replacing the adjoint action with its push-forward differential $\rho_*$:

$$\mathcal{PT}_{s\to t}^\rho = \mathcal{T}\exp\left(-\int_s^t \rho_*\left(\mathrm{ad}_{\xi(\tau)}^\dagger\right)d\tau\right), \quad s \le t. \quad (8)$$

**Role of the Parallel Transport.** Conceptually, the operator $\mathcal{PT}_{s\to t}^\rho$ (red) in Figure 1 provides a mechanism for transporting a vector $\mathbf{x}_s \in \mathbb{R}^d$ at time $s$ along a trajectory induced by a geodesic flow $\xi(t)$ on $G$, thereby yielding its corresponding vector $\mathbf{x}_t \in \mathbb{R}^d$ at time $t$. Geometrically, parallel transport specifies how to move a vector along a curve while keeping its direction. On a curved manifold, this rule is path dependent: if we start with a tangent vector at $A$ and parallel transport it successively along the geodesic sides $A \to B \to C \to A$ on the sphere, the vector returns rotated relative to its initial orientation. This rotation measures the curvature enclosed by the triangle $ABC$. Our parallel transport operator $\mathcal{PT}_{s\to t}^\rho$ is designed to encode exactly this effect, accumulating the holonomy generated by the learned geodesic flow along the trajectory.

**Time-reversal of Lie-geodesics.** Given a geodesic curve $g(t)$ associated with (7), we introduce its reversed-time parameterization defined by $g^\sharp(t) := g(T-t)$, for $t \in [0, T]$. Under this reparameterization, the associated left–trivialized velocity transforms as

$$\xi^\sharp(t) = -\xi(T-t), \quad \dot{\xi}^\sharp(t) = -\mathrm{ad}_{\xi^\sharp(t)}^\dagger \xi^\sharp(t), \quad (9)$$

*Table 1.* Structure Constants and Group Representations of Various Lie Algebras

| Lie Group | Structure Constants $\mathbb{C}_{ij,kl}^{pq}$ | | | Differential $\rho_*(X)$ |
|---|---|---|---|---|
| GL($V$), $\mathfrak{gl}(V)$ | $\mathbb{C}_{ij,kl}^{il} = \delta_{jk}$, | $\mathbb{C}_{ij,kl}^{kj} = -\delta_{il}$ | | $\rho_*(X) = X$ |
| O($V, B$), $\mathfrak{o}(V, B)$ | $\mathbb{C}_{ij,kl}^{il} = \delta_{jk}$, | $\mathbb{C}_{ij,kl}^{kj} = -\delta_{il}$, | $\mathbb{C}_{ij,kl}^{qm} = -B_{ik}B^{mq}\delta_{jl} + B_{il}B^{mq}\delta_{jk}$ | $\rho_*(X) = BX, \quad X^\top B + BX = 0$ |

indicating that $g^\sharp$ describes the same geodesic trajectory but traversed in the opposite time direction. Proposition B.4 suggests that the parallel transport along the reversed geodesic $g^\sharp$ can be correspondingly defined by

$$\frac{d}{dt}\mathcal{PT}_{s\to t}^{\rho,\sharp} = -\rho_*\left(\mathrm{ad}_{\xi^\sharp(t)}^\dagger\right)\mathcal{PT}_{s\to t}^{\rho,\sharp}, \ \mathcal{PT}_{s\to s}^{\rho,\sharp} = \mathrm{Id}. \quad (10)$$

This construction satisfies a natural consistency condition between forward and reversed-time transports given by $(\mathcal{PT}_{s\to t}^{\rho})^{-1} = \mathcal{PT}_{T-t\to T-s}^{\rho,\sharp}$ (blue) in Figure 1. In this geometric framework, the standard Koopman-structured dynamics given by (3) naturally extend into a set of forward-reverse dynamics $(\mathbf{x}_t, \mathbf{x}_s^\sharp)$ that incorporates curvature-induced structure. Specifically, we refer to this generalized dynamical formulation as the *Euler–Poincaré Neural Dynamics*:

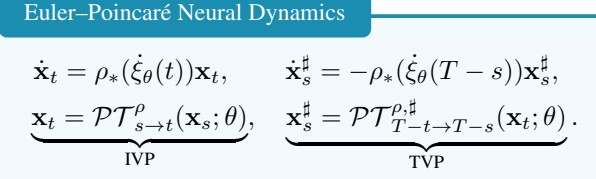

Euler–Poincaré Neural Dynamics

$$\dot{\mathbf{x}}_t = \rho_*(\dot{\xi}_\theta(t))\mathbf{x}_t, \qquad \dot{\mathbf{x}}_s^\sharp = -\rho_*(\dot{\xi}_\theta(T-s))\mathbf{x}_s^\sharp,$$

$$\underbrace{\mathbf{x}_t = \mathcal{PT}_{s\to t}^\rho(\mathbf{x}_s; \theta)}_{\text{IVP}}, \qquad \underbrace{\mathbf{x}_s^\sharp = \mathcal{PT}_{T-t\to T-s}^{\rho,\sharp}(\mathbf{x}_t; \theta)}_{\text{TVP}}.$$

This newly suggested formulation naturally leads us to generalize the original learning objective in (4) by incorporating the intrinsic geometric parallel transport operator:

$$\mathcal{J}(\theta) := \mathbb{E}_{\hat{\mathbf{x}}\sim\mathbb{P}_{\text{data}}}\left[\int_{[0,T]}\|\mathcal{PT}_{0\to t}^\rho(\hat{\mathbf{x}}_0; \theta) - \hat{\mathbf{x}}_t\|^2 dt\right]. \quad (11)$$

With this objective, we jointly handle the *initial value problem* (IVP) and the *terminal value problem* (TVP) in a single reversible framework by exploiting the property in (9) of the Lie-geodesic transport, thus enabling endpoint-conditioned TVP reconstruction at test time using the IVP-trained parameters without any retraining. To fully realize this unified approach, our next step is to develop a principled neural parameterization of parallel transport in (21) and propose an efficient, reversible computational scheme that seamlessly solves both IVP and TVP.

## 3. Parameterization: Neural Geodesic Fields

This section introduces a neural parameterization framework for geodesics $\xi_\theta$ and the parallel transport operator $\mathcal{PT}^\rho(\cdot; \theta)$. This approach directly integrates geometric constraints into learnable models by formulating geodesic flows explicitly on Lie groups. To concretely realize this formulation, we first establish an explicit coordinate representation.

For a basis $\mathbb{B} = \{E_a\}_{a=1}^r$ of Lie algebra $\mathfrak{g}$ with *structure constants* $[E_i, E_j] = \sum_{k=1}^r \mathbb{C}_{ij}^k E_k$, let $\langle\cdot, \cdot\rangle_\mathfrak{g}$ be a positive

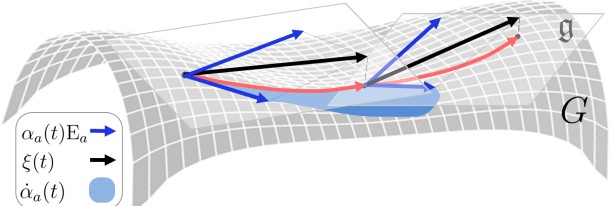

*Figure 2.* **Neural Geodesic Field.** The Lie algebra coefficients $\alpha(t)$ provide a coordinate representation of the left-trivialized velocity $\xi(t)$ in the tangent space, with $\dot{\alpha}(t)$ governing its evolution according to the Euler–Arnold equation, and the geodesic $g(t)$ is reconstructed from $\xi(t)$ to transport the state.

definite inner product on $\mathfrak{g}$ and write its Riemannian Gram matrix and inverse by $\mathbf{G}_{ab} := \langle E_a, E_b\rangle_\mathfrak{g}$, $\mathbf{G}^{ab} := (\mathbf{G}^{-1})^{ab}$. For a smooth curve $g(t) \in G$, the left–trivialized velocity $\xi$ can be expressed as a linear combination of the basis elements $\{E_a\}$ with time–dependent coefficients that represent the coordinates of $\dot{g}(t)$ in the moving tangent frame:

$$\xi(t) := g(t)^{-1}\dot{g}(t) = \sum_{a=1}^r \alpha_a(t)E_a \in \mathfrak{g}.$$

Here, $\alpha_a(t)$ encodes the projection of the body velocity onto the Lie algebra basis, and the collection $\alpha_a(t)$ thus forms the coordinate representation of the velocity field in the intrinsic frame attached to the group element $g(t)$. By exploiting this coordinate representation, we parameterize the vector fields $\alpha_a$ to ensure that the underlying Lie-group geodesic dynamics strictly follow the Euler–Arnold equation. Specifically, the parameterization is structured through neural networks, leading to *neural geodesic fields*:

**Proposition 3.1** (Neural Geodesic Fields)**.** *Let $\xi_\theta(t) := \xi(t; \theta) := \sum_{a=1}^{|\mathrm{Dim}(\mathbb{B})|} \alpha_a(t, z; \theta)E_a$ be parameterized velocity fields on Lie algebra $\mathfrak{g}$ called **neural geodesic fields**. Then, the coefficient ordinary equation in the following solves the geodesic equation $\nabla_{\dot{g}}^\mathfrak{g}\dot{g} = 0$ in (7):*

$$\dot{\alpha}_a = -\sum_{i,b,c,d}^{|\mathrm{Dim}(\mathbb{B})|} \alpha_i(t; \theta)\alpha_b(t; \theta)\mathbf{G}^{ac}\mathbb{C}_{ic}^d\mathbf{G}_{db}, \quad (12)$$

Following the construction in Proposition 3.1, proposed Euler–Poincaré neural dynamics can be repurposed as:

$$\dot{\mathbf{x}}_t = \rho_*\left(\sum_{a=1}\dot{\alpha}_a(t, z; \theta)E_a\right)\mathbf{x}_t. \quad (13)$$

Here, the group representation $\rho$ determines the induced algebraic structure and the associated set of structure constants $\mathbb{C}$. Table 1 summarizes the explicit structure constants

$\mathbb{C}_{ij}^k$ and the corresponding differential representations $\rho_*$ for the classical Lie algebras, providing the algebraic backbone that governs the evolution of the neural geodesic fields introduced in Proposition 3.1.

**Conditioning in Neural Geodesic Fields.** In the proposed formulation, the neural coefficients $\alpha_a(t, z; \theta)$ are conditioned on the boundary configuration variable $z$, which specifies the context of the geodesic problem. For an IVP, the conditioning variable is the initial state $z = \mathbf{x}_0$, ensuring that the neural geodesic field evolves forward in time from the starting configuration. Conversely, for a TVP, the conditioning variable becomes the terminal state $z = \mathbf{x}_T$, enabling backward-time transport toward the target configuration. This unified conditioning scheme allows the same neural parameterization $\xi_\theta(t, z)$ to seamlessly represent both forward and reverse geodesic flows.

## 4. Parallelization: Euler–Poincaré Prefix Scan

In long-horizon geometric dynamical systems, sequential Lie-group updates and curvature interactions render standard discretizations prohibitively expensive in runtime and memory. We introduce an extended algebraic framework that preserves geometric structure while enabling scalable parallel computation.

**Extended Lie Algebra.** Let $\mathfrak{g}$ be a real Lie algebra and write the exterior square $\mathfrak{g} \wedge \mathfrak{g} := \mathrm{span}\{X \wedge Y : X, Y \in \mathfrak{g}\}$ with $X \wedge Y := -Y \wedge X$ and bilinearity in each slot. Define the vector space $\mathfrak{G} := \mathfrak{g} \oplus (\mathfrak{g} \wedge \mathfrak{g})$ and endow it with the binary operation $\star$ given by

$$(\mathrm{U}, \mathrm{C}) \star (\mathrm{U}', \mathrm{C}') := \left(\mathrm{U} + \mathrm{U}', \mathrm{C} + \mathrm{C}' + \frac{1}{2}\mathrm{U} \wedge \mathrm{U}'\right), \quad (14)$$

for any pairs $\mathrm{U}, \mathrm{U}' \in \mathfrak{g}, \mathrm{C}, \mathrm{C}' \in \mathfrak{g} \wedge \mathfrak{g}$. Then $(\mathfrak{G}, \star)$ is a new Lie group. We now recall the notion of a *semidirect product*, *i.e.*, $\ltimes$, which generalizes the concept of a direct product by incorporating a group action. We first introduce a smooth group action $G \times \mathfrak{G} \to \mathfrak{G}$ as $(g, (\mathrm{U}, C)) \mapsto g \curvearrowright (\mathrm{U}, C)$:

$$g \curvearrowright (\mathrm{U}, C) := (\mathrm{Ad}_g \mathrm{U}, \mathrm{Ad}_g C), \quad g = \exp(\mathrm{U}), \quad (15)$$

where $\mathrm{Ad}_G[X] = G[X]G^{-1}$ denotes the adjoint representation of $G$ acting on its Lie algebra $\mathfrak{g}$ and on its wedge product extension $\mathfrak{g} \wedge \mathfrak{g}$, *i.e.*, $\mathrm{Ad}_g[X \wedge Y] = (\mathrm{Ad}_g[X]) \wedge (\mathrm{Ad}_g[Y])$. Using this action, we define the semidirect product of $G$ and $\mathfrak{G}$ referred to as *parallel transport group* $\mathfrak{H}_{\mathrm{PT}}$:

---

**Definition 4.1** (Parallel Transport Group). For a set of pairs $X = (\mathrm{U}, \mathrm{C})$ and $Y = (\mathrm{U}', \mathrm{C}')$, we introduce $\mathfrak{H}_{\mathrm{PT}} := G \ltimes \mathfrak{G} \ni (g, X)$, equipped with group multiplication rule given explicitly by semidirect product:

$$(g, X) \ltimes (h, Y) = (gh, \mathrm{X} \star (g \curvearrowright Y))$$
$$= \left(gh, \mathrm{U} + \mathrm{Ad}_g \mathrm{U}', \mathrm{C} + \mathrm{Ad}_g \mathrm{C}' + \tfrac{1}{2}\mathrm{U} \wedge \mathrm{Ad}_g \mathrm{U}'\right).$$

where $(\star, \curvearrowright)$ is defined in (14) and (15), respectively.

---

Section B.2 provides the corresponding exact local coordinate together with generalized mathematical backgrounds underlying this construction. Since $\mathfrak{H}_{\mathrm{PT}}$ is a *group* under the semidirect product operation, its multiplication is *associative* by definition. Therefore, for any elements $(g, X), (h, \mathrm{Y}), (f, \mathrm{Z}) \in \mathfrak{H}_{\mathrm{PT}}$, the following relation holds:

$$(g, \mathrm{X}) \ltimes [(h, Y) \ltimes (f, Z)] = [(g, \mathrm{X}) \ltimes (h, \mathrm{Y})] \ltimes (f, \mathrm{Z}).$$

Having established the extended Lie algebra and its semidirect product, in what follows, we introduce a discrete formulation of parallel transport and demonstrate how algebraic associativity naturally supports scalable computation.

**Discrete Parallel Transport.** Let $0 = t_0 < t_1 < \cdots < t_M = T$ partition the time interval $[0, T]$, and consider a piecewise $C^1$ velocity field $\xi : [0, T] \to \mathfrak{g}$. At each discrete time step, we approximate the continuous parallel transport by evaluating $\xi$ at the left endpoint of the interval and scaling by the step size $h_m := t_m - t_{m-1}$. We then exponentiate these Lie algebra increments to obtain discrete Lie group elements $g_m$. Initially, we set the second-order curvature correction $C_m$ to zero. Each increment is lifted to the previously defined extended structure $\mathfrak{H}_{\mathrm{PT}}$, yielding:

$$\mathrm{U}_m := h_m \xi(t_{m-1}), \quad g_m := \exp(\mathrm{U}_m), \quad \mathrm{C}_m := 0 \in \mathfrak{g} \wedge \mathfrak{g},$$

and lift $\Delta S_m := (g_m, (\mathrm{U}_m, \mathrm{C}_m)) \in \mathfrak{H}_{\mathrm{PT}}$. For the initialized cumulative state $\mathcal{S}_0 := (e, (0, 0)) \in \mathfrak{H}_{\mathrm{PT}}$, we recursively define the sequence of cumulative geometric states:

$$\mathcal{S}_m := \mathcal{S}_{m-1} \ltimes \Delta\mathrm{S}_m = (\mathcal{G}_m, (\mathcal{U}_m, \mathcal{C}_m)), \quad 1 \le m \le M,$$

This semidirect-product recursion provides explicit component-wise updates for the cumulative state and makes clear how each increment $\Delta S_m$ contributes to the global geometric evolution. By accumulating both group motion and curvature $(\mathcal{U}_m, \mathcal{C}_m)$ in an associative way, it defines a discrete parallel-transport operator that captures higher-order non-commutative effects while remaining compatible with parallel prefix scans:

---

**Discretization of Parallel Transport**

$$\mathcal{G}_m = \mathcal{G}_{m-1} g_m, \qquad \Phi(\mathrm{X} \wedge \mathrm{Y}) = \tfrac{1}{2}[\mathrm{X}, \mathrm{Y}],$$
$$\mathcal{U}_m = \mathcal{U}_{m-1} + \mathrm{Ad}_{\mathcal{G}_{m-1}} \mathrm{U}_m,$$
$$\mathcal{C}_m = \mathcal{C}_{m-1} + \mathrm{Ad}_{\mathcal{G}_{m-1}} \mathrm{C}_m + \frac{1}{2}\mathcal{U}_{m-1} \wedge \mathrm{Ad}_{\mathcal{G}_{m-1}} \mathrm{U}_m,$$
$$\widehat{\mathcal{PT}}_{0 \to t_m}^{\rho} = \rho(\mathcal{G}_m) \exp\left(\rho_*\big(\Phi(\mathcal{C}_m)\big)\right),$$
$$\widehat{\mathcal{PT}}_{t_m \to 0}^{\rho, \sharp} = \exp\left(-\rho_*\big(\Phi(\mathcal{C}_m)\big)\right)\rho(\mathcal{G}_m^{-1}).$$

---

The cumulative Lie group element $\mathcal{G}_m$ approximates the geometric flow, while the cumulative algebraic quantities $\mathcal{U}_m$ and $\mathcal{C}_m$ encode intrinsic geometric features. Ultimately,

these discrete parallel transport operators allow us to explicitly propagate states forward and backward in time through the discretized evolution:

$$\mathbf{x}_m = \widehat{\mathcal{PT}}^{\rho}_{0 \to t_m}(\mathbf{x}_0; \theta), \quad \mathbf{x}_0 = \widehat{\mathcal{PT}}^{\rho, \sharp}_{t_m \to 0}(\mathbf{x}_m; \theta), \quad (16)$$

forming the computational basis for efficient and geometrically faithful numerical integration of dynamical systems. As a concrete illustration of this forward transport mechanism, we provide a two-step matrix example in Appendix F. Proposition B.6 further details how each recursive algebraic accumulation $(\mathcal{U}_m, \mathcal{C}_m)$ can be consistently parameterized by the neural geodesic coefficients $\alpha_a(t, z; \theta)$, providing a differentiable bridge between the discrete transport algebra and learnable Lie–group dynamics. We now quantify the theoretical deviation between the proposed discrete parallel transport operator and the exact continuous counterpart:

---

**Proposition 4.2** (Informal). *For the time interval $[t_m, t_{m+1}]$ and $h = t_{m+1} - t_m$. For a sufficiently small step size $h$ and compact Lie group $G$, we have operator norm $\| \cdot \|_{\mathrm{op}}$ in (33), which is upper bounded as follows:*

$$\left\| \widehat{\mathcal{PT}}^{\rho}_{0 \to t_m} - \mathcal{PT}^{\rho}_{0 \to t_m} \right\|_{op} \lesssim \mathcal{O}\left( M h^2 \sqrt{\kappa_{\max}} \mathrm{Diam}(G) \right)$$

$$+ \mathcal{O}\left( M h^2 \left( \|\dot{\xi}\|_{\max} + \frac{\mathrm{Diam}(G)^2}{h^2} \right) \right),$$

*where $\kappa_{\max}$ and $\mathrm{Diam}(G)$ denote maximal sectional curvature and diameter of $G$ and we set $\|\dot{\xi}\|_{\max} := \max_{k \leq M} \|\dot{\xi}_k\|$.*

---

Proposition 4.2 demonstrates that when the underlying dynamics exhibit fast changes (*e.g.*, $\|\dot{\xi}\| \uparrow$) or when the curvature of the manifold is large (*e.g.*, $\kappa_{max} \uparrow$), smaller integration steps are required (*e.g.*, $h \downarrow$) to maintain geometric fidelity and prevent the accumulation of geometric distortion over time.

**Euler–Poincaré Prefix Scan.** The discrete construction introduced above not only provides a theoretically rigorous approximation to continuous geometric evolutions but also yields notable computational benefits arising directly from its intrinsic algebraic structure. Importantly, the semidirect product operation $\ltimes$ satisfies the essential algebraic property of *associativity*, thus enabling highly efficient computational strategies via the well-established **prefix parallel scan** algorithm Blelloch (1989). By explicitly exploiting this associativity, cumulative states $\mathcal{S}_m$ can be flexibly regrouped and rearranged for any $1 \leq m \leq M$:

$$\mathcal{S}_m = \Delta\mathrm{S}_0 \cdots \ltimes (\Delta\mathrm{S}_{m-2} \ltimes \Delta\mathrm{S}_{m-1}) \ltimes \Delta\mathrm{S}_m \cdots \ltimes \Delta\mathrm{S}_M$$

$$= \Delta\mathrm{S}_0 \cdots \ltimes \Delta\mathrm{S}_{m-2} \ltimes (\Delta\mathrm{S}_{m-1} \ltimes \Delta\mathrm{S}_m) \cdots \ltimes \Delta\mathrm{S}_M.$$

This iterative parallel scan procedure is illustrated in Figure 1, *(right)* red. Building upon the geometric mechanics formulation outlined above, we define the resulting parallel scan algorithm constructed via semidirect product operations on the discrete parallel transport operators as the *Euler–Poincaré prefix scan*. The reverse formulation complements the forward Euler–Poincaré prefix scan by propagating the cumulative state in the opposite temporal direction, depicted in Figure 1, *(right)* blue.

$$\mathcal{S}^{\sharp}_m = \Delta\mathrm{S}_M \ltimes \cdots \ltimes (\Delta\mathrm{S}_{m+1} \ltimes \Delta\mathrm{S}_m) \ltimes \Delta\mathrm{S}_{m-1} \ltimes \cdots$$

$$= \Delta\mathrm{S}_M \ltimes \cdots \ltimes \Delta\mathrm{S}_{m+1} \ltimes (\Delta\mathrm{S}_m \ltimes \Delta\mathrm{S}_{m-1}) \ltimes \cdots .$$

This construction represents the discrete realization of the time-reversed parallel transport operator $\mathcal{PT}^{\rho, \sharp}_{T \to t}$ satisfying $(\mathcal{PT}^{\rho}_{0 \to t})^{-1} = \mathcal{PT}^{\rho, \sharp}_{T \to T-t}$. By applying the same associative semidirect product $\ltimes$ in reverse order, the cumulative geometric state $\mathcal{S}^{\sharp}_m$ encodes inverse mapping of the forward transport. The complete description of the scanning procedure is provided in Section C.

## 5. Related Work

Multiple Koopman-based methods approximate nonlinear dynamics in lifted observable spaces by learning finite-dimensional linear operators. Koopa Liu et al. (2023), KooNPro Zheng et al. (2025), KoNODE Bai & Ding (2025), and MamKO LI et al. (2025) extend this framework by incorporating non-stationarity, slow–fast temporal separation, spectral variability, and streaming or nonuniform observations into the operator-learning process. KoVAE Naiman et al. (2024) further advances this line of research by enforcing linear dynamics within a latent generative model, with the imposed linear prior constraining latent temporal evolution and promoting coherent trajectory structure. Despite these developments, operator evolution in all such approaches is defined in extrinsic or learned coordinate systems, and long-horizon stability consequently depends on the degree to which the learned linearizations faithfully approximate the underlying nonlinear dynamics. A distinct research direction embeds geometric structure directly into model architectures. Symplectic neural formulations such as SympNets Jin et al. (2020) and SymODEN Zhong et al. (2020) preserve invariants through structured parameterizations of vector fields or energy functions, resulting in improved long-term integration accuracy for conservative systems. Recent generative trajectory models, including EqMotion Xu et al. (2023) and GeoTDM Han et al. (2024), incorporate Euclidean or rigid-body symmetries, thereby providing physically grounded inductive biases that enhance the modeling of complex dynamical processes.

## 6. Experiment

In this section, we evaluate the proposed Euler–Poincaré Neural Dynamics (EPND) as a surrogate model for molecular time evolution. The experiments evaluate whether EPND maintains numerical stability over long rollouts in protein folding and electrolyte hydration systems, assess whether

*Table 2.* **Protein Folding Dynamics.** Runtime efficiency and prediction accuracy on protein folding trajectories.

| Method | Villin (35-residue) | | | | TRP-CAGE (20-residue) | | | |
|---|---|---|---|---|---|---|---|---|
| | Inference (s) | Train (s) | Best MSE ↓ | $C_\alpha$-RMSD (Å) ↓ | Inference (s) | Train (s) | Best MSE ↓ | $C_\alpha$-RMSD (Å) ↓ |
| Koopa Liu et al. (2023) | $3.74 \times 10^{-2}$ | $9.29 \times 10^{-2}$ | $2.32 \times 10^{-2}$ | $1.51 \times 10^0$ | $3.15 \times 10^{-2}$ | $9.16 \times 10^{-2}$ | $4.75 \times 10^{-2}$ | $3.56 \times 10^0$ |
| KoVAE Naiman et al. (2024) | $5.34 \times 10^{-2}$ | $7.76 \times 10^{-1}$ | $8.16 \times 10^{-2}$ | $1.03 \times 10^0$ | $6.55 \times 10^{-2}$ | $9.49 \times 10^{-1}$ | $7.13 \times 10^{-2}$ | $2.95 \times 10^0$ |
| KooNPro Zheng et al. (2025) | $8.57 \times 10^{-2}$ | $1.83 \times 10^{-1}$ | $8.98 \times 10^{-2}$ | $1.63 \times 10^0$ | $6.52 \times 10^{-2}$ | $1.43 \times 10^{-1}$ | $7.81 \times 10^{-2}$ | $\mathbf{2.81 \times 10^0}$ |
| KoNODE Bai & Ding (2025) | $5.69 \times 10^{-1}$ | $4.53 \times 10^0$ | $1.08 \times 10^0$ | $1.57 \times 10^1$ | $5.62 \times 10^{-1}$ | $4.78 \times 10^0$ | $1.03 \times 10^{-1}$ | $5.28 \times 10^0$ |
| MamKO LI et al. (2025) | $1.27 \times 10^{-1}$ | $3.17 \times 10^0$ | $3.30 \times 10^{-2}$ | $1.80 \times 10^0$ | $2.24 \times 10^{-1}$ | $3.29 \times 10^0$ | $6.95 \times 10^{-2}$ | $4.15 \times 10^0$ |
| SympNets Jin et al. (2020) | $2.87 \times 10^{-1}$ | $9.29 \times 10^{-1}$ | $2.18 \times 10^0$ | $3.10 \times 10^0$ | $2.88 \times 10^{-1}$ | $9.04 \times 10^{-1}$ | $3.07 \times 10^0$ | $4.12 \times 10^0$ |
| SymODEN Zhong et al. (2020) | $1.51 \times 10^1$ | $7.25 \times 10^{-1}$ | $2.01 \times 10^1$ | $7.35 \times 10^0$ | $1.38 \times 10^1$ | $6.25 \times 10^{-1}$ | $3.02 \times 10^0$ | $4.22 \times 10^0$ |
| EqMotion Xu et al. (2023) | $4.81 \times 10^0$ | $4.79 \times 10^0$ | $8.02 \times 10^1$ | $2.01 \times 10^1$ | $6.56 \times 10^0$ | $6.62 \times 10^0$ | $5.02 \times 10^2$ | $2.84 \times 10^1$ |
| GeoTDM Han et al. (2024) | $2.66 \times 10^2$ | $8.21 \times 10^{-1}$ | $5.23 \times 10^{-2}$ | $1.32 \times 10^0$ | $2.46 \times 10^2$ | $7.69 \times 10^{-1}$ | $3.70 \times 10^{-2}$ | $3.00 \times 10^0$ |
| Ours | $\mathbf{2.61 \times 10^{-2}}$ | $\mathbf{5.33 \times 10^{-2}}$ | $\mathbf{5.56 \times 10^{-3}}$ | $\mathbf{5.09 \times 10^{-1}}$ | $\mathbf{1.87 \times 10^{-2}}$ | $\mathbf{4.30 \times 10^{-2}}$ | $\mathbf{7.60 \times 10^{-3}}$ | $2.91 \times 10^0$ |

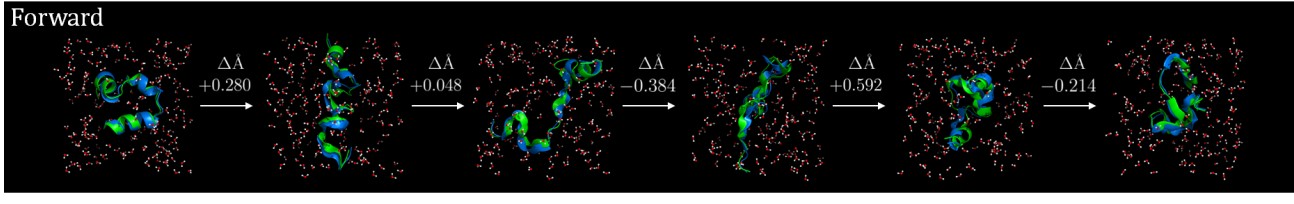

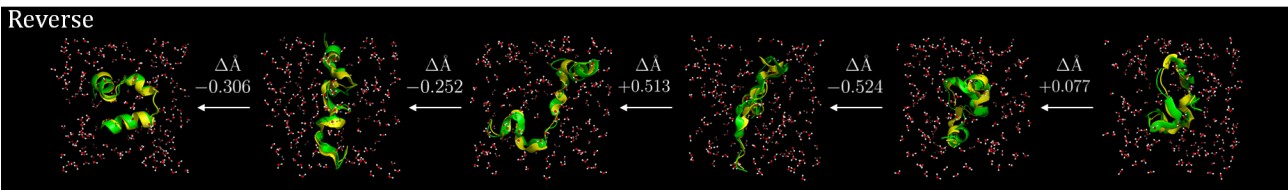

*Figure 3.* **Visualization of Protein Folding Dynamics.** Each snapshot shows the time evolution with $M = 1.0 \times 10^5$ in total, corresponds to 10ns in real time. Green protein figure depicts the ground truth, with blue and orange trajectories are made with IVP/TVP respectively.

the learned geometric flow preserves the structural information necessary for reconstructing earlier configurations from terminal states, and quantify the computation time advantages of the Euler–Poincaré prefix scan under fixed rollout lengths relative to Koopman-based and generative surrogate models. We consider three representative atomistic systems with two protein dynamics, and the hydration shell of $Mg^{2+}$.

*Villin Headpiece* and *TRP–CAGE* are fast-folding peptide benchmarks commonly used to study protein dynamics. Villin is a 35-residue peptide containing 582 atoms including hydrogens Dobson (2003); Lindorff-Larsen et al. (2011); Chiu et al. (2005); Nagel et al. (2023), while TRP–CAGE is a 20-residue miniprotein comprising 284 atoms including hydrogens Neidigh et al. (2002); Qiu et al. (2002); Marinelli et al. (2009). We also consider an *electrolyte hydration system* to examine ion-centered dynamics. Specifically, we model the aqueous hydration of $Mg^{2+}$, which forms a tightly ordered first shell of six coordinating water molecules, a feature commonly seen in diverse RNA folding Teplova et al. (2020), ion channel selectivity Takeda et al. (2014), and nucleic acid interactions Leonarski et al. (2017).

All evaluations are performed on three molecular benchmarks. The models are trained and tested under a shared protocol using rollout comparison in $M = 1000$ rollout settings, evaluation for long-horizon prediction, and reverse

prediction by TVP are conducted with our model. We generate atomistic molecular dynamics datasets for protein folding and electrolyte hydration systems. All simulations employ explicit solvent and all-atom force fields, with further details of simulation settings, trajectory generation, and preprocessing are provided in Section D.

We compare EPND with Koopman-based and geometric surrogate models under a shared rollout horizon. Since several baselines exceed error tolerances or exhaust single GPU memory for long horizon, model comparisons are reported at a unified rollout length of 1000 steps. Performance is evaluated using best MSE, $C_\alpha$-RMSD for protein systems, and wall clock runtime for inference and training. As shown in Fig. 6, the computational complexity differs across model families. GeoTDM uses transformer-based dynamics with complexity $O(M^2)$ while KoNODE requires numerical ODE integration with complexity $O(M)$ and the Euler–Poincaré prefix scan used in EPND performs updates in $O(\log M)$. Additional implementation details are provided in Section D.

**Model Comparison.** Tab. 2 summarizes the best MSE and $C_\alpha$-RMSD for all benchmarks. EPND achieves the lowest MSE across villin, TRP-CAGE, and $Mg^{2+}$ hydration at rollout length 1000, with corresponding $C_\alpha$-RMSD values of $5.09 \times 10^{-1}$ Å for villin and 2.91 Å for TRP-CAGE, and a best MSE of $3.03 \times 10^{-2}$ and lowest $Mg - O$ ion length

| Method | Inference (s) | Train (s) | Best MSE ↓ | $\Delta(Mg - O)$ |
|---|---|---|---|---|
| Koopa Liu et al. (2023) | $6.44 \times 10^{-2}$ | $1.31 \times 10^{-1}$ | $4.87 \times 10^{-1}$ | $1.97 \times 10^{0}$ |
| KoVAE Naiman et al. (2024) | $6.87 \times 10^{-2}$ | $9.51 \times 10^{-1}$ | $3.61 \times 10^{-2}$ | $1.09 \times 10^{-1}$ |
| KooNPro Zheng et al. (2025) | $9.35 \times 10^{-2}$ | $1.56 \times 10^{-1}$ | $4.25 \times 10^{-2}$ | $6.50 \times 10^{-1}$ |
| KoNODE Bai & Ding (2025) | $5.77 \times 10^{-1}$ | $4.65 \times 10^{0}$ | $5.56 \times 10^{-1}$ | $3.37 \times 10^{2}$ |
| MamKO LI et al. (2025) | $1.07 \times 10^{-1}$ | $1.46 \times 10^{0}$ | $4.33 \times 10^{-2}$ | $2.19 \times 10^{0}$ |
| SympNets Jin et al. (2020) | $2.72 \times 10^{-1}$ | $8.91 \times 10^{-1}$ | $7.22 \times 10^{-2}$ | $1.55 \times 10^{0}$ |
| SymODEN Zhong et al. (2020) | $1.43 \times 10^{1}$ | $6.35 \times 10^{-1}$ | $3.03 \times 10^{0}$ | $1.78 \times 10^{1}$ |
| EqMotion Xu et al. (2023) | $4.62 \times 10^{0}$ | $4.74 \times 10^{0}$ | $7.99 \times 10^{1}$ | $3.27 \times 10^{2}$ |
| GeoTDM Han et al. (2024) | $1.67 \times 10^{2}$ | $4.75 \times 10^{-1}$ | $3.16 \times 10^{-1}$ | $3.55 \times 10^{0}$ |
| Ours | $\mathbf{1.59 \times 10^{-2}}$ | $\mathbf{5.25 \times 10^{-2}}$ | $\mathbf{3.03 \times 10^{-2}}$ | $\mathbf{9.28 \times 10^{-2}}$ |

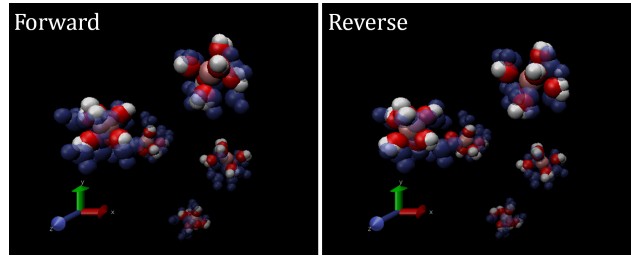

*Figure 4.* **Quantitative Results and Visualization of Electrolyte Hydration Dynamics.** Left table shows the comparison of runtime and prediction accuracy on electrolyte hydration dynamics. Right figure shows the visualization with transparent blue figure depicts the ground truth, while the opaque figure shows IVP/TVP respectively.

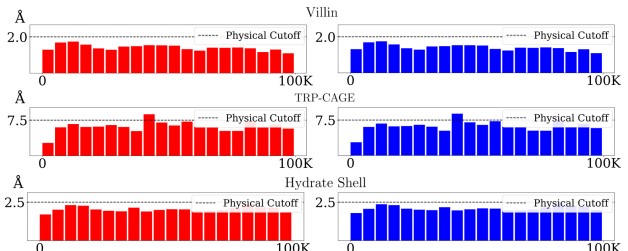

*Figure 5.* **IVP/TVP Comparison.** The graph shows the RMSD for $10^5$ timesteps. The colorized graphs in both columns correspond to trajectories of IVP and TVP, respectively.

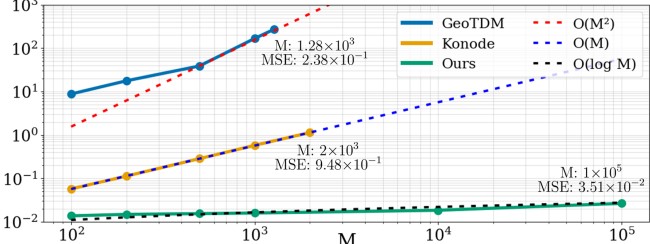

*Figure 6.* **Temporal Complexity Comparison.** The graph shows the evaluation time shift along the timestep with log scale.

difference of $9.28 \times 10^{-2}$ Å for the hydration system. These results indicate that EPND predicts both peptide structures and ion coordinated water configurations with improved accuracy. Fig. 4 compares 1000 step rollouts with reference trajectories. EPND maintains close alignment with ground truth, while Koopman-based and generative baselines exhibit increasing deviation. The inference time is between 1.5 and 5 times faster than Koopman-based baselines under equal rollout lengths, with training time between 1.5 and 6 times faster, both gains arising from the logarithmic update structure of the Euler-Poincaré prefix scan.

**Long-Horizon Prediction in IVP.** Long-horizon stability is assessed by integrating each model for $10^5$ steps in an open-loop initial value setting. This evaluation protocol is particularly stringent, as autonomous rollout without corrective feedback is known to accentuate the accumulation of local truncation errors, often leading to structural drift or qualitative changes in the underlying dynamics. In the context of biomolecular motion, where physically plausible trajectories require the preservation of global geometric relations over extended timescales, such long-horizon assessments provide a direct measure of whether a learned dynamical model captures the intrinsic flow geometry. Fig. 5 (left) shows that EPND maintains a stable forward evolution without drift or divergence, with $C_\alpha$-RMSD curves remaining nearly constant for the entire rollout. Best MSE records $4.63 \times 10^{-2}$ for villin, $6.64 \times 10^{-2}$ for TRP-CAGE, and $3.51 \times 10^{-2}$ for $Mg^{2+}$ hydration. These results indicate that the Lie algebraic update mitigates stepwise error accumulation and supports long-horizon propagation.

**Backward Prediction in TVP.** We assess the invertibility of the learned flow by formulating a terminal value problem (TVP). Given a terminal observation $\mathbf{x}_T$, we evaluate whether the model admits a stable inverse by estimating an initial state that satisfies $\mathbf{x}_0 \approx \mathcal{PT}_{T \to 0}^{\rho, \sharp}(\hat{\mathbf{x}}_T; \theta)$, which ensures consistency with reference $\hat{\mathbf{x}}_0$. This requirement of inferring an initial state from a terminal observation mirrors the structure of many scientific inverse problems such as pharmacokinetic reverse-dosimetry (Tan et al., 2007), atmospheric source reconstruction (De Meutter et al., 2018), and cosmological inference from present-day data (Shallue & Eisenstein, 2023). After training, all parameters remain fixed and the model is interpreted as a deterministic geometric flow. Fig. 3 shows that the recovered initial structures remain close to the reference, with reconstruction errors within $1.0 \times 10^{-3}$ of the forward best MSE. Backward rollouts in Fig. 5 confirm that EPND maintains a stable reverse evolution. Together, these results demonstrate that the parallel-transport formulation preserves the geometric structure required for stable inversion over long trajectories.

# 7. Conclusion

We introduced Euler–Poincaré Neural Dynamics (EPND), a geometric-mechanics framework that models dynamical evolution through neural geodesic fields and parallel transport on Lie groups, enforcing the Euler–Arnold equation to encode symmetry, curvature, and reversibility. To support scalable computation, we developed the parallel transport group and the Euler–Poincaré prefix scan, an associative semidirect-product formulation for efficient forward and backward composition of discrete geometric updates. Ex-

periments on protein folding and electrolyte hydration show that EPND improves accuracy, stability, and speed over existing surrogates, while enabling terminal-value inference and suggesting applicability to broader physical systems.

## Impact Statement

This paper presents work whose goal is to advance the field of machine learning. Our work has many potential societal consequences, none of which must be specifically highlighted here.

## Acknowledgments

This work was supported by the Institute of Information & Communications Technology Planning & Evaluation(IITP)-ICT Creative Consilience Program grant funded by the Korea government(MSIT)(IITP-2026-RS-2020-II201819). This work was also supported by the National Research Foundation of Korea(NRF) grant funded by the Korea government(MSIT)(RS-2026-25495269). This research was supported by Basic Science Research Program through the NRF funded by the Korea government(MSIT)(RS-2026-25522655). Additional support was provided by Korea University Faculty Start-up Research Grant(K2506311). Further support was provided by Korea University Faculty Start-up Research Grant(K2506541).

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

## A. Notation Table

This section consolidates the symbols and conventions employed throughout the paper to enhance clarity.

| Symbol | Meaning |
| --- | --- |
| $x(t)$ | Trajectory solving $\dot{x}(t) = v(t, x(t))$ with initial $x(0) = x_0$ |
| $v(t, x)$ | Time–dependent vector field on $X$ |
| $\Phi_{s,t} : X \to X$ | Flow map transporting $x(s)$ to $x(t)$; $\Phi_{s,t} \circ \Phi_{r,s} = \Phi_{r,t}$ |
| $K_{s,t}$ | Koopman operator $K_{s,t} f = f \circ \Phi_{s,t}$, $K_{s,t} K_{r,s} = K_{r,t}$ |
| $L_t$ | Koopman generator $(L_t f)(x) = \nabla f(x) \cdot v(t, x)$ |
| $\iota(x)$ | Identity observable $\iota(x) = x$ |
| $\mathfrak{g} = T_e G$ | Lie algebra of $G$ with Lie bracket $[\cdot, \cdot]$ |
| $B = \{E_a\}_{a=1}^r$ | Fixed basis of $\mathfrak{g}$, $[E_i, E_j] = \sum_k C_{ij}^k E_k$ |
| $\mathbb{C}_{ij}^k$ | Structure constants in the basis $B$ |
| $\langle \cdot, \cdot \rangle_{\mathfrak{g}}$ | Inner product on $\mathfrak{g}$; Gram $G_{ab}$, inverse $G^{ab}$ |
| $L_g$ | Left translation on $G$, $L_g(h) = gh$ |
| $\mathrm{Ad}_g$ | Adjoint action on $\mathfrak{g}$, $\mathrm{Ad}_g(X) = gXg^{-1}$ |
| $\mathrm{ad}_\xi$ | Infinitesimal adjoint, $\mathrm{ad}_\xi(\eta) = [\xi, \eta]$ |
| $\mathrm{ad}_\xi^\dagger$ | Metric adjoint, $\langle \mathrm{ad}_\xi^\dagger u, v \rangle_{\mathfrak{g}} = \langle u, [\xi, v] \rangle_{\mathfrak{g}}$ |
| $I : \mathfrak{g} \to \mathfrak{g}^*$ | Inertia operator; momentum $m = \delta\ell/\delta\xi = I\xi$ |
| $\nabla_g$ | Levi–Civita connection of a left–invariant metric on $G$ |
| $g(t), \xi(t)$ | Curve $g(t) \in G$ and left–trivialized velocity $\xi(t) = g(t)^{-1}\dot{g}(t) \in \mathfrak{g}$ |
| *Geodesic eq.* | $\dot{\xi} = -\mathrm{ad}_\xi^\dagger \xi$, equivalently $\nabla_g \dot{g}\dot{g} = 0$ |
| $V \simeq \mathbb{R}^d$ | Representation space |
| $\rho : G \to \mathrm{GL}(V)$ | (Linear) group representation |
| $\rho^* : \mathfrak{g} \to \mathfrak{gl}(V)$ | Differential of $\rho$ at the identity |
| $\mathrm{PT}_{s \to t}^\rho$ | Parallel transport on $V$ along $g(\cdot)$: $\mathrm{PT}_{s \to t}^\rho = \mathcal{T}\exp\left(-\int_s^t \rho^*(\mathrm{ad}_{\xi(\tau)}^\dagger)d\tau\right)$ |
| $\mathrm{PT}_{s \to t}^{\rho, \sharp}$ | Time–reversed transport; $(\mathrm{PT}_{s \to t}^\rho)^{-1} = \mathrm{PT}_{t \to s}^{\rho, \sharp}$ |
| $\alpha_a(t, z; \theta)$ | Neural coefficients conditioned on boundary $z \in X$ |
| $\xi_\theta(t)$ | Neural geodesic field $\xi_\theta(t) = \sum_a \alpha_a(t, z; \theta)E_a$ |
| $\dot{\alpha}_a$ | Coefficient ODE: $\dot{\alpha}_a = -\sum_{i,b,c,d} \alpha_i \alpha_b G_{ac} C_{ic}^d G_{db}$ |
| $X \wedge Y$ | Exterior product on $\mathfrak{g}$, $X \wedge Y = -Y \wedge X$ |
| $\Lambda(\mathfrak{g})$ | Exterior algebra; truncation $\Lambda^{\leq R}(\mathfrak{g}) = \bigoplus_{r=1}^R \mathfrak{g}^{\wedge r}$ |
| $\mathcal{G}$ | Extended Lie algebra $\mathfrak{g} \oplus (\mathfrak{g} \wedge \mathfrak{g})$ |
| $(U, C) \star (U', C')$ | Graded product on $\mathcal{G}$: $(U + U', C + C' + \frac{1}{2}U \wedge U')$ |
| $g \curvearrowright (U, C)$ | Group action $(\mathrm{Ad}_g U, \mathrm{Ad}_g C)$ |
| $H_{\mathrm{PT}}$ | Parallel transport group $G \ltimes \mathcal{G}$ |
| $H_{\mathrm{PT}}^{\leq R}$ | Graded extension $G \ltimes \Lambda^{\leq R}(\mathfrak{g})$ with components $Z^{(r)} \in \mathfrak{g}^{\wedge r}$ |
| $0 = t_0 < \cdots < t_M = T$ | Time partition with steps $h_m = t_m - t_{m-1}$ |
| $\Delta U_m$ | Step increment $\Delta U_m = h_m \xi(t_{m-1})$ |
| $g_m$ | Incremental group element $g_m = \exp(\Delta U_m)$ |
| $\Delta S_m$ | Lifted increment in $H_{\mathrm{PT}}$: $\Delta S_m = (g_m, (\Delta U_m, 0))$ |
| $S_m$ | Cumulative state in $H_{\mathrm{PT}}$: $S_m = S_{m-1} \ltimes \Delta S_m = (G_m, (U_m, C_m))$ |
| $G_m$ | Cumulative group product, $G_m = G_{m-1}g_m$ |
| $U_m, C_m$ | First/second–order accumulators; $\Phi(X \wedge Y) = \frac{1}{2}[X, Y]$ |
| $\mathrm{PT}_{0 \to t_m}^{\rho, \mathrm{d}}$ | Discrete transport: $\mathrm{PT}_{0 \to t_m}^{\rho, \mathrm{d}} = \rho(G_m)\exp\left(\rho^*(\Phi(C_m))\right)$ |
| $\ltimes$ | Semidirect product enabling associative parallel prefix scan |
| $\rho(G) \subset \mathrm{GL}(V)$ | General linear case |
| $\rho(G) \subset O(V, B)$ | Orthogonal case with bilinear form $B$; $X^\top B + BX = 0$ for $\rho^*(X)$ |
| $\rho(G) \subset \mathrm{Sp}(V, \Omega)$ | Symplectic case with form $\Omega$; $X^\top \Omega + \Omega X = 0$ for $\rho^*(X)$ |
| $\| \cdot \|_{\mathrm{op}}$ | Operator norm on $V$ |
| $\kappa_{\max}$ | Upper bound of sectional curvature of $G$ |
| $\mathrm{Diam}(G)$ | Diameter of $G$ under the chosen metric |
| $O(\cdot)$ | Big–O asymptotic order in step size and geometric terms |

## B. Mathematical Backgrounds and Proofs

In this appendix, we collect essential mathematical preliminaries and present detailed proofs necessary to fully understand and rigorously verify the theoretical results established in the main body of the paper. We aim to make the presentation self-contained, providing explicit definitions, identities, and intermediate derivations.

### B.1. Connection-Frame Geometry of Lie Group

We first introduce the mathematical structures required to precisely characterize the geometry underlying our theoretical framework. The concepts of left-invariant frames, Maurer–Cartan forms, and Levi–Civita connections on Lie groups are reviewed here to set a solid foundation for the proofs that follow.

**Elementary Lie Groups.** A *Lie group* $G$ is a smooth manifold endowed with a smooth group operation $(g, h) \mapsto g \cdot h$ and a smooth inversion map $g \mapsto g^{-1}$. The identity element is denoted by $e \in G$. Associated with every Lie group is its *Lie algebra* $\mathfrak{g}$, defined as the tangent space at the identity, $\mathfrak{g} = T_e G$. The Lie algebra encodes infinitesimal motions of the group through a bilinear and antisymmetric operation called the *Lie bracket*, $[X, Y] \in \mathfrak{g}$ for $X, Y \in \mathfrak{g}$, which satisfies the Jacobi identity

$$[X, [Y, Z]] + [Y, [Z, X]] + [Z, [X, Y]] = 0, \quad \forall X, Y, Z \in \mathfrak{g}.$$

The *exponential map* establishes a local correspondence between the Lie algebra and the Lie group. It is defined as $\exp : \mathfrak{g} \to G$, $X \mapsto \exp(X)$, and describes how infinitesimal algebra elements generate finite group displacements via one-parameter subgroups. Closely related is the *adjoint representation*, $\mathrm{Ad}_g : \mathfrak{g} \to \mathfrak{g}$, which expresses how group elements act on their algebra through conjugation,

$$\mathrm{Ad}_g(X) = \frac{d}{dt}\Big|_{t=0} g \exp(tX) g^{-1}, \qquad g \in G, \ X \in \mathfrak{g},$$

and satisfies $\mathrm{Ad}_{gh} = \mathrm{Ad}_g \circ \mathrm{Ad}_h$. To represent higher-order geometric quantities such as curvature or holonomy, we employ the *wedge product* (also called the *exterior product*). For any vector space $V$, this operation is an alternating multilinear map satisfying $X \wedge Y = -Y \wedge X$. Repeated application defines antisymmetric tensor spaces $V^{\wedge r}$, constructed as $V^{\wedge r} = \underbrace{V \wedge V \wedge \cdots \wedge V}_{r \text{ times}}$, and their direct sum forms the *exterior algebra*:

$$\Lambda(V) = V \oplus (V \wedge V) \oplus (V \wedge V \wedge V) \oplus \cdots.$$

For the computational feasibility, this hierarchy is often truncated at finite order $R$, leading to the *truncated exterior algebra*

$$\Lambda(V)^{\leq R} = V \oplus (V \wedge V) \oplus (V \wedge V \wedge V) \oplus \cdots \oplus (V^{\wedge R}),$$

which retains all multilinear interactions up to degree $R$ while discarding higher-order terms.

**Left-invariant Lie Algebra.** Let $\{E_a\}_{a=1}^r$ be a basis of the Lie algebra $\mathfrak{g}$ with $[E_i, E_c] = \sum_d \mathbb{C}_{ic}^d E_d$. Let $\langle \cdot, \cdot \rangle_\mathfrak{g}$ be an inner product with Gram matrix $\mathbf{G}_{ab} = \langle E_a, E_b \rangle_\mathfrak{g}$ and inverse $\mathbf{G}^{ab}$. We define the left-invariant frame and Maurer–Cartan coframe by $e_a(g) = (L_g)_* E_a$ and $\theta^a(g) = (L_{g^{-1}})^* E^a$, where $\{E^a\}$ is the dual basis, and moving frame is given as $\{e_i^a\}$. The left-invariant metric is given as

$$g = \sum_{a,b} \mathbf{G}_{ab} \theta^a \otimes \theta^b = \sum_{a,b} \mathbf{G}_{ab} \theta^a \otimes \theta^b.$$

where left translations are isometries: $(L_h)^* g = g$ for any transformation $h \in G$. In this setting, the coframe and frame are left-invariant: $(L_h)^* \theta^a = \theta^a$ and $(L_h)_* e_a = e_a \circ L_h$. In local coordinates $x = (x^1, \ldots, x^n)$, let us write two coordinate systems as follows:

$$\theta^a = \sum_i e_i^a(x) dx^i = \sum_i e_i^a(x) dx^i, \qquad e_a = \sum_i \mathrm{E}_a^i(x) \partial_i = \sum_i \mathrm{E}_a^i(x) \partial_i,$$

and set the frame as matrix expressions $e(x) = [e_i^a(x)]$ where basis can be presented as $\mathrm{E}(x) = [\mathrm{E}_a^i(x)] = (e(x)^{-1})^\top$. Then the Riemannian Gram matrix and its inverse in the coordinate basis can be written as

$$g_{ij}(x) = \sum_{a,b} e_i^a(x) \mathbf{G}_{ab} e_j^b(x) = \left( e(x)^\top \mathbf{G} e(x) \right)_{ij} = \sum_{a,b} e_i^a(x) \mathbf{G}_{ab} e_j^b(x), \tag{17}$$

$$g^{ij}(x) = \sum_{a,b} \mathrm{E}_a^i(x) \mathbf{G}^{ab} \mathrm{E}_b^j(x) = \left( \mathrm{E}(x) \mathbf{G}^{-1} \mathrm{E}(x)^\top \right)^{ij} = \sum_{a,b} \mathrm{E}_a^i(x) \mathbf{G}^{ab} \mathrm{E}_b^j(x). \tag{18}$$

**Proposition B.1** (Koszul–adjoint identity in a left-invariant frame). *Let $e_a = e_a^i \partial_i$ be a left-invariant moving frame on a Lie group $G$ equipped with a left-invariant metric $g = \mathbf{G}_{ab}\theta^a \otimes \theta^b$, and let $X = X^c e_c$, $Y = Y^b e_b$ be vector fields expressed in this frame. Then the Levi–Civita connection satisfies*

$$\nabla_X^{\mathfrak{g}} Y = \left(X^i \partial_i Y^k + X^i Y^j \Gamma_{ij}^k\right)\partial_k = \tfrac{1}{2}\left([X,Y] - \mathrm{ad}_X^\dagger Y - \mathrm{ad}_Y^\dagger X\right),$$

*where the adjoint transpose $\mathrm{ad}^\dagger$ and covariant derivative components are given by*

$$\left(\mathrm{ad}_{e_c}^\dagger e_b\right)^a = \mathbf{G}^{ad}\mathbb{C}_{dcb} = \mathbf{G}^{ad}\mathbf{G}_{dm}\mathbb{C}_{cb}^m = \mathbb{C}_{cb}^a, \quad \left(\nabla_{e_c}^{\mathfrak{g}} e_b\right)^a = \tfrac{1}{2}\left(\mathbb{C}_{cb}^a - \mathbf{G}^{ad}\mathbb{C}_{dcb} - \mathbf{G}^{ad}\mathbb{C}_{dbc}\right).$$

*Proof.* We start with the Levi–Civita symbols in a coordinate basis $\{\partial_i\}$,

$$\Gamma_{ij}^k(x) = \tfrac{1}{2}g^{k\ell}(x)\left(\partial_i g_{j\ell}(x) + \partial_j g_{i\ell}(x) - \partial_\ell g_{ij}(x)\right),$$

and pass to a moving frame $e_a = \mathrm{E}_a^i \partial_i$ with dual coframe $\theta^a = e_i^a dx^i$, where $\mathrm{E}_a^i e_j^a = \delta_j^i$ and $e_i^a \mathrm{E}_b^i = \delta_b^a$. The connection one–forms in this frame are $\omega_b^a = \Gamma_{bc}^a \theta^c$ so that $\nabla_{e_c} e_b = \Gamma_{bc}^a e_a$. By writing the Lie brackets as $[e_b, e_c] = \mathbb{C}_{bc}^d e_d$ and lowering the first index with the metric $\mathbf{G}_{ad} = \langle e_a, e_d\rangle$ gives $\mathbb{C}_{abc} := \mathbf{G}_{ad}\mathbb{C}_{bc}^d = \langle e_a, [e_b, e_c]\rangle$. Since $g(e_b, e_c) = \mathbf{G}_{bc}$ is constant for a left-invariant metric, the derivative terms in the Koszul formula vanish and one obtains, for left-invariant vector fields $X, Y, Z$,

$$2\langle \nabla_X^{\mathfrak{g}} Y, Z\rangle = \langle [X,Y], Z\rangle - \langle [Y,Z], X\rangle - \langle [X,Z], Y\rangle,$$

We specialize Koszul formula to the unit frame $X = e_c$, $Y = e_b$, $Z = e_a$ to obtain metric components

$$2\Gamma_{abc} := 2\langle \nabla_{e_c} e_b, e_a\rangle = \mathbb{C}_{cba} - \mathbb{C}_{bac} - \mathbb{C}_{cab} = \mathbb{C}_{acb} - \mathbb{C}_{cba} - \mathbb{C}_{bca}. \tag{19}$$

This shows that the Christoffel symbol can be read as $\Gamma_{abc} = \tfrac{1}{2}\left(\mathbb{C}_{cba} - \mathbb{C}_{bac} + \mathbb{C}_{acb}\right)$ where $\Gamma_{bc}^a = \mathbf{G}^{ad}\Gamma_{dbc}$. This gives rise to definition of $\mathrm{ad}^\dagger$ in an explicit form,

$$\left(\mathrm{ad}_{e_c}^\dagger e_b\right)^a = \mathbf{G}^{ad}\mathbb{C}_{bcd}, \qquad \left(\nabla_{e_c}^{\mathfrak{g}} e_b\right)^a = \frac{1}{2}\left(\mathbb{C}_{cb}^a - \mathbf{G}^{ad}\mathbb{C}_{bcd} - \mathbf{G}^{ad}\mathbb{C}_{cbd}\right),$$

which agrees with $\Gamma_{bc}^a = \mathbf{G}^{ad}\Gamma_{dbc}$ above. Next, we aim to verify the main statement: The coordinate expression of the Levi–Civita connection $\nabla_X^{\mathfrak{g}} Y = X^i \partial_i Y^k + X^i Y^j \Gamma_{ij}^k = \tfrac{1}{2}\left([X,Y] - \mathrm{ad}_X^\dagger Y - \mathrm{ad}_Y^\dagger X\right)$ is consistent with the Koszul formula in the case of left-invariant vector fields on a Lie group with a left-invariant metric. Let $X = X^i e_i$, $Y = Y^j e_j$ be arbitrary vector fields in the left-invariant frame $\{e_i^a\}$. The definition of Lie bracket gives

$$[X,Y]^k = (X^i \partial_i Y^j e_j - Y^j \partial_j X^i e_i + X^i Y^j [e_i, e_j])^k = X^i \partial_i Y^k - Y^i \partial_i X^k + X^i Y^j \mathbb{C}_{ij}^k,$$

Now consider the adjoint transpose with respect to vector fields $X$ and $Y$:

$$\mathrm{ad}_{e_c}^\dagger e_b = \mathbf{G}^{ad}\mathbb{C}_{bcd} e_a \quad \Rightarrow \quad (\mathrm{ad}_X^\dagger Y)^k = X^i Y^j \mathbf{G}^{ka}\mathbb{C}_{jia}, \quad (\mathrm{ad}_Y^\dagger X)^k = Y^i X^j \mathbf{G}^{ka}\mathbb{C}_{ija}$$

We now compute the right-hand side of the Koszul expression:

$$\frac{1}{2}\left([X,Y] - \mathrm{ad}_X^\dagger Y - \mathrm{ad}_Y^\dagger X\right)^k = \frac{1}{2}\left(X^i \partial_i Y^k - Y^i \partial_i X^k + X^i Y^j \mathbb{C}_{ij}^k - X^i Y^j \mathbf{G}^{ka}\mathbb{C}_{jia} - Y^i X^j \mathbf{G}^{ka}\mathbb{C}_{ija}\right)$$

$$= X^i \partial_i Y^k - \tfrac{1}{2}Y^i \partial_i X^k + \tfrac{1}{2}X^i Y^j \left(\mathbb{C}_{ij}^k - \mathbf{G}^{ka}\mathbb{C}_{jia} - \mathbf{G}^{ka}\mathbb{C}_{ija}\right).$$

Using the musical isomorphism $\mathbb{C}_{ij}^k = \mathbf{G}^{ka}\mathbb{C}_{aij}$, one can substitute structure constant so that:

$$\mathbb{C}_{ij}^k - \mathbf{G}^{ka}\mathbb{C}_{jia} - \mathbf{G}^{ka}\mathbb{C}_{ija} = \mathbf{G}^{ka}(\mathbb{C}_{aij} - \mathbb{C}_{jia} - \mathbb{C}_{ija}).$$

On the other hand, from the earlier frame-based expression for the connection defined in (19), one can rewrite the following $\Gamma_{ij}^k = \tfrac{1}{2}\mathbf{G}^{ka}(\mathbb{C}_{aij} - \mathbb{C}_{jia} - \mathbb{C}_{ija})$ by using the antisymmetry $\mathbb{C}_{iaj} = -\mathbb{C}_{ija}$, which matches the previous expression. Hence, we conclude that

$$X^i Y^j \Gamma_{ij}^k = \frac{1}{2}X^i Y^j \left(\mathbb{C}_{ij}^k - \mathbf{G}^{ka}\mathbb{C}_{jia} - \mathbf{G}^{ka}\mathbb{C}_{ija}\right),$$

and thus the two formulations of the Levi–Civita connection agree:

$$\nabla_X^{\mathfrak{g}} Y = X^i \partial_i Y^k + X^i Y^j \Gamma_{ij}^k = \frac{1}{2}\left([X,Y] - \mathrm{ad}_X^\dagger Y - \mathrm{ad}_Y^\dagger X\right).$$

$\square$

---

**Proposition B.2** (Geodesic equation in a Lie group with left-invariant metric). *Let $g(t) \in G$ be a smooth curve on a Lie group $G$ equipped with a left-invariant Riemannian metric, and define the left-trivialized velocity $\xi(t) := (L_{g(t)}^{-1})_* \dot{g}(t) \in \mathfrak{g}$. Then $g(t)$ is a geodesic if and only if*

$$(L_{g(t)}^{-1})_*\left(\nabla_{\dot{g}}^{\mathfrak{g}} \dot{g}\right) = \dot{\xi} + \mathrm{ad}_\xi^\dagger \xi = 0. \tag{20}$$

---

*Proof.* We start from the Koszul–adjoint identity established in Proposition B.1:

$$\nabla_X^{\mathfrak{g}} Y = \frac{1}{2}\left([X,Y] - \mathrm{ad}_X^\dagger Y - \mathrm{ad}_Y^\dagger X\right),$$

which holds for all left-invariant vector fields $X, Y \in \mathfrak{X}(G)$ under a left-invariant metric. Now, let $g(t)$ be a smooth curve on $G$ with tangent vector $\dot{g}(t) \in T_{g(t)}G$. Since the Lie bracket is antisymmetric, we have $[\dot{g}, \dot{g}] = 0$. substituting $X = Y = \dot{g}$ into the above identity gives

$$\nabla_{\dot{g}}^{\mathfrak{g}} \dot{g} = \frac{1}{2}\left([\dot{g}, \dot{g}] - \mathrm{ad}_{\dot{g}}^\dagger \dot{g} - \mathrm{ad}_{\dot{g}}^\dagger \dot{g}\right) = -\mathrm{ad}_{\dot{g}}^\dagger \dot{g}.$$

Next, recall that the velocity of the curve in the left-trivialized form is given by $\xi(t) := (L_{g(t)}^{-1})_* \dot{g}(t) \in \mathfrak{g}$, so that $\dot{g}(t) = (L_{g(t)})_* \xi(t)$. Because the connection is left-invariant, we have

$$(L_{g(t)}^{-1})_*\left(\nabla_{\dot{g}}^{\mathfrak{g}} \dot{g}\right) = \frac{d}{dt}\left((L_{g(t)}^{-1})_* \dot{g}\right) + (L_{g(t)}^{-1})_*\left(\mathrm{ad}_{\dot{g}}^\dagger \dot{g}\right)$$
$$= \dot{\xi} + \mathrm{ad}_\xi^\dagger \xi$$

The first term gives $\dot{\xi}$, and using $\dot{g} = (L_{g(t)})_* \xi$ in the second term yields the result. The geodesic condition $\nabla_{\dot{g}}^{\mathfrak{g}} \dot{g} = 0$ is thus equivalent to $\dot{\xi} + \mathrm{ad}_\xi^\dagger \xi = 0$. making the geodesic equation on the Lie algebra $\mathfrak{g}$ under the left-invariant metric. $\square$

---

**Proposition B.3** (Parallel transport under a group representation). *Suppose the group acts linearly on a finite–dimensional vector space $V \simeq \mathbb{R}^d$ via a smooth representation $\rho : G \to GL(V)$ with differential $\rho_* : \mathfrak{g} \to \mathfrak{gl}(V)$. Then the parallel transport of any vector $v(s) \in V$ along the trajectory $g(t)$ is governed by the linear ODE*

$$\frac{d}{dt} v(t) = -\rho_*\left(\mathrm{ad}_{\xi(t)}^\dagger\right) v(t), \qquad v(s) = v_0 \in V,$$

*whose solution defines the parallel transport operator $\mathcal{PT}_{s\to t}^\rho : V \to V$, given explicitly by the time–ordered exponential*

$$\mathcal{PT}_{s\to t}^\rho = \mathcal{T}\exp\left(-\int_s^t \rho_*\left(\mathrm{ad}_{\xi(\tau)}^\dagger\right) d\tau\right), \quad s \le t. \tag{21}$$

---

*Proof.* Let $g(t) \in G$ be a smooth geodesic curve with left-trivialized velocity $\xi(t) := (L_{g(t)}^{-1})_* \dot{g}(t) \in \mathfrak{g}$. Let $\rho : G \to GL(V)$ be a smooth group representation on a finite-dimensional vector space $V$, with induced Lie algebra representation $\rho_* : \mathfrak{g} \to \mathfrak{gl}(V)$. Let $v(t) \in V$ be a time-dependent vector, interpreted as the representation-space counterpart of a vector field $U(t) \in T_{g(t)}G$ along the curve, via the induced representation $\rho$. Our goal is to define a notion of parallel transport of $v(t)$ along $g(t)$ such that it mirrors the covariant derivative condition

$$\nabla_{\dot{g}}^{\mathfrak{g}} U = 0 \quad \Longleftrightarrow \quad \frac{D}{dt}\left((L_{g(t)}^{-1})_* U(t)\right) = -\mathrm{ad}_{\xi(t)}^\dagger u(t).$$

For the simplicity, let us define time-dependent vector fields $A(t) := \rho_*\left(\mathrm{ad}^\dagger_{\xi(t)}\right) \in \mathfrak{gl}(V)$. Since $t \mapsto \xi(t)$ is continuous along the geodesic and both $\mathrm{ad}^\dagger : \mathfrak{g} \to \mathrm{End}(\mathfrak{g})$ and $\rho_* : \mathfrak{g} \to \mathfrak{gl}(V)$ are linear and continuous, the map $t \mapsto A(t)$ is continuous on any compact interval $[s, t]$. Next, consider the initial value problem

$$\dot{v}(\tau) = -A(\tau)v(\tau), \qquad v(s) = v_0 \in V.$$

By Picard–Lindelöf theorem Teschl (2012), there exists a unique solution on $[s, t]$. Equivalently, $v$ satisfies the Volterra equation $v(t) = v_0 - \int_s^t A(\tau)v(\tau)d\tau$. Next, we iterate this identity to yield the infinite series so-called the *Peano–Baker series* Baake & Schlaegel (2011) given by

$$v(t) = \left( I - \int_s^t A(\tau_1)d\tau_1 + \int_s^t \int_s^{\tau_1} A(\tau_1)A(\tau_2)d\tau_2 d\tau_1 - \cdots \right) v_0.$$

Let $M := \sup_{\tau \in [s,t]} \|A(\tau)\|$. The $n$-th term has maximal norm bounded by the term $M^n(t-s)^n/n!$, hence the infinite series converges absolutely and uniformly, defining a bounded linear operator $\Phi$:

$$\Phi(t, s) := \mathcal{T} \exp\left( \int_s^t A(\tau)d\tau \right) := I + \sum_{n=1}^\infty (-1)^n \int_{s \leq \tau_n \leq \cdots \leq \tau_1 \leq t} A(\tau_1) \cdots A(\tau_n) d\tau_n \cdots d\tau_1, \quad \Phi(t, s) \in \mathrm{End}(V),$$

Uniform convergence permits differentiation termwise; using the fundamental theorem of calculus for nested integrals one obtains $\frac{d}{dt}\Phi(t, s) = -A(t)\Phi(t, s)$ where $\Phi(s, s) = I$. This demonstrates that $v(t) = \Phi(t, s)v_0$ solves the ODE, and uniqueness implies that every solution is of this form. By definition, the series for $\Phi(t, s)$ is the time-ordered exponential, namely

$$\Phi(t, s) = \mathcal{T} \exp\left( -\int_s^t A(\tau)d\tau \right) = \mathcal{T} \exp\left( -\int_s^t \rho_*\left(\mathrm{ad}^\dagger_{\xi(\tau)}\right)d\tau \right).$$

Setting $\mathcal{PT}^\rho_{s \to t} := \Phi(t, s)$ gives the stated formula, and the composition and inverse properties follow from uniqueness of the Cauchy problem, so $\mathcal{PT}^\rho_{s \to t}$ is the unique parallel transport operator on $V$ along $g(\cdot)$. $\qquad\square$

---

**Proposition B.4** (Reversed Parallel Transport). *Let $\rho : G \to \mathrm{End}(V)$ be a smooth representation and $\xi^\sharp : [s, t] \to \mathfrak{g}$ a time-dependent Lie algebra curve. Define the* reversed parallel transport *operator $\mathcal{PT}^{\rho,\sharp}_{s \to t} : V \to V$ by*

$$\frac{d}{dt}\mathcal{PT}^{\rho,\sharp}_{s \to t} = -\rho_*\left(\mathrm{ad}^\dagger_{\xi^\sharp(t)}\right)\mathcal{PT}^{\rho,\sharp}_{s \to t}, \qquad \mathcal{PT}^{\rho,\sharp}_{s \to s} = \mathrm{Id}. \tag{22}$$

*Then $\mathcal{PT}^{\rho,\sharp}_{s \to t}$ transports elements of $V$ backward along the adjoint-covariant flow generated by $\xi^\sharp(t)$. Equivalently, it satisfies*

$$\mathcal{PT}^{\rho,\sharp}_{s \to t} = \left(\mathcal{PT}^\rho_{t \to s}\right)^{-1}, \qquad \frac{d}{dt}\mathcal{PT}^\rho_{t \to s} = \rho_*\left(\mathrm{ad}^\dagger_{\xi^\sharp(t)}\right)\mathcal{PT}^\rho_{t \to s}.$$

*Hence the negative sign in (22) encodes transport in the reverse time direction, propagating dual or covariant quantities under the adjoint-dual dynamics. If the inner product on $\mathfrak{g}$ is bi-invariant (i.e. $\mathrm{ad}^\dagger_\xi = -\mathrm{ad}_\xi$), the evolution reduces to*

$$\frac{d}{dt}\mathcal{PT}^{\rho,\sharp}_{s \to t} = \rho_*(\mathrm{ad}_{\xi^\sharp(t)})\mathcal{PT}^{\rho,\sharp}_{s \to t},$$

*corresponding to classical right-invariant Lie transport.*

---

*Proof.* As similar to previous proof, let us consider time-dependent vector fields $A(t) := \rho_*\left(\mathrm{ad}^\dagger_{\xi(t)}\right) \in \mathfrak{gl}(V)$. Since $t \mapsto A(t)$ is continuous on compact intervals, the linear operator ODE $\frac{d}{dt}\Phi(t, s) = -A(t)\Phi(t, s)$ with $\Phi(s, s) = \mathrm{Id}$ admits again a unique fundamental solution given equivalently by the Peano–Baker series:

$$\Phi(t, s) = \mathcal{T} \exp\left( -\int_s^t A(\tau)d\tau \right) = \mathrm{Id} + \sum_{n=1}^\infty (-1)^n \int_{s \leq \tau_n \leq \cdots \leq \tau_1 \leq t} A(\tau_1) \cdots A(\tau_n) d\tau_n \cdots d\tau_1.$$

For the reversed transport expressed by $\mathcal{PT}^{\rho,\sharp}_{s\to t} := \Phi(t,s)$, the proposed construction gives a form of differential equation which is exactly (22):

$$\frac{d}{dt}\mathcal{PT}^{\rho,\sharp}_{s\to t} = -A(t)\mathcal{PT}^{\rho,\sharp}_{s\to t}, \qquad \mathcal{PT}^{\rho,\sharp}_{s\to s} = \mathrm{Id},$$

Recall that Liouville's formula gives

$$\frac{d}{dt}\log\det\Phi(t,s) = \mathrm{tr}\big(-A(t)\big), \qquad \det\Phi(t,s) = \exp\Big(-\int_s^t \mathrm{tr}\, A(\tau)d\tau\Big) \neq 0,$$

so $\mathcal{PT}^{\rho,\sharp}_{s\to t}$ is invertible for all $t$. To compare with the forward transport, let $\Psi^{\mathrm{L}}(t,s)$ and $\Psi^{\mathrm{R}}(t,s)$ denote the left- and right-ordered fundamental solutions:

$$\frac{d}{dt}\Psi^{\mathrm{L}}(t,s) = A(t)\Psi^{\mathrm{L}}(t,s), \qquad \frac{d}{dt}\Psi^{\mathrm{R}}(t,s) = \Psi^{\mathrm{R}}(t,s)A(t), \qquad \Psi^{\mathrm{L}}(s,s) = \Psi^{\mathrm{R}}(s,s) = \mathrm{Id}.$$

Then

$$\frac{d}{dt}\big(\Psi^{\mathrm{R}}(t,s)\big)^{-1} = -A(t)\big(\Psi^{\mathrm{R}}(t,s)\big)^{-1}, \qquad \frac{d}{dt}\big(\Psi^{\mathrm{L}}(t,s)\big)^{-1} = -\big(\Psi^{\mathrm{L}}(t,s)\big)^{-1}A(t).$$

If we take the forward transport to be $\mathcal{PT}^{\rho}_{t\to s} := \Psi^{\mathrm{L}}(t,s)$ (as written in the lemma), then

$$\frac{d}{dt}\mathcal{PT}^{\rho}_{t\to s} = A(t)\mathcal{PT}^{\rho}_{t\to s}, \qquad \mathcal{PT}^{\rho,\sharp}_{s\to t} = \big(\mathcal{PT}^{\rho}_{t\to s}\big)^{-1},$$

which establishes the equivalence statement in the lemma. Alternatively, defining the forward transport by the right-ordered convention makes the reversed equation appear with the generator on the left exactly as in (22). For any $v_s \in V$, the curve $v(t) := \mathcal{PT}^{\rho,\sharp}_{s\to t}v_s$ solves $\frac{d}{dt}v(t) = -A(t)v(t)$, hence it evolves backward relative to the forward flow driven by $A(t)$. This observation leads to intertwining identity

$$\mathcal{PT}^{\rho}_{t\to s}v(t) = \mathcal{PT}^{\rho}_{t\to s}\mathcal{PT}^{\rho,\sharp}_{s\to t}v_s = v_s,$$

which confirms the backward-transport interpretation. The transport also satisfies the cocycle property

$$\mathcal{PT}^{\rho,\sharp}_{s\to u} = \mathcal{PT}^{\rho,\sharp}_{t\to u}\mathcal{PT}^{\rho,\sharp}_{s\to t}, \qquad s \leq t \leq u,$$

which follows either from uniqueness of the IVP or directly from the time-ordered exponential representation. Finally, if the inner product on $\mathfrak{g}$ is $\mathrm{Ad}$-invariant, then $\mathrm{ad}^{\dagger}_{\xi} = -\mathrm{ad}_{\xi}$ and therefore $A(t) = -\rho_*(\mathrm{ad}_{\xi^{\sharp}(t)})$, yielding the specialization

$$\frac{d}{dt}\mathcal{PT}^{\rho,\sharp}_{s\to t} = \rho_*\big(\mathrm{ad}_{\xi^{\sharp}(t)}\big)\mathcal{PT}^{\rho,\sharp}_{s\to t},$$

which is the classical right-invariant Lie transport form. All assertions in the lemma follow. $\qquad\square$

---

*Proposition* 3.1 (Neural Geodesic Fields). Let $\xi_\theta(t) := \xi(t;\theta) := \sum_{a=1}^{|\mathrm{Dim}(\mathbb{B})|}\alpha_a(t,z;\theta)\mathrm{E}_a$ be parameterized velocity fields on Lie algebra $\mathfrak{g}$ called a **neural geodesic fields**. Then, the coefficient ordinary equation given by

$$\dot{\alpha}_a = -\sum_{i,b,c,d}^{|\mathrm{Dim}(\mathbb{B})|}\alpha_i(t;\theta)\alpha_b(t;\theta)\mathbf{G}^{ac}\mathbb{C}^d_{ic}\mathbf{G}_{db}, \tag{23}$$

solves the geodesic equation $\nabla^{\mathfrak{g}}_{\dot{g}}\dot{g} = 0$ in (7).

---

*Proof.* Write the left–trivialized velocity in the fixed algebra basis as $\xi_\theta(t) = \sum_a \alpha_a(t,z;\theta)\mathrm{E}_a$. Following by Proposition B.2, the geodesic equation under a left–invariant metric is $\dot{\xi} + \mathrm{ad}^{\dagger}_{\xi}\xi = 0$. We expand both terms in the basis to derive

$$\dot{\xi} = \sum_a \dot{\alpha}_a\mathrm{E}_a, \qquad \mathrm{ad}^{\dagger}_{\xi}\xi = \sum_{i,b}\alpha_i\alpha_b\,\mathrm{ad}^{\dagger}_{\mathrm{E}_i}\mathrm{E}_b.$$

This shows that the geodesic equation $\nabla^{\mathfrak{g}}_{\dot{g}} \dot{g} = 0$ in this expression is equivalent to

$$\sum_a \dot{\alpha}_a \mathrm{E}_a + \sum_{i,b} \alpha_i \alpha_b \, \mathrm{ad}^{\dagger}_{\mathrm{E}_i} \mathrm{E}_b = 0.$$

It remains to express the components of $\mathrm{ad}^{\dagger}$. With structure constants $[\mathrm{E}_i, \mathrm{E}_c] = \sum_d \mathbb{C}^d_{ic} \mathrm{E}_d$ and Gram matrix $\mathbf{G}_{ab} = \langle \mathrm{E}_a, \mathrm{E}_b \rangle_{\mathfrak{g}}$, the adjoint transpose is characterized by

$$\langle \mathrm{ad}^{\dagger}_{\mathrm{E}_i} \mathrm{E}_b, \mathrm{E}_a \rangle_{\mathfrak{g}} = \langle \mathrm{E}_b, [\mathrm{E}_i, \mathrm{E}_a] \rangle_{\mathfrak{g}} = \sum_d \mathbf{G}_{bd} \mathbb{C}^d_{ia}.$$

Raising the index with by using Riemannian Gram matrix $\mathbf{G}^{ac}$, one obtains $\left( \mathrm{ad}^{\dagger}_{\mathrm{E}_i} \mathrm{E}_b \right)^a = \sum_{c,d} \mathbf{G}^{ac} \mathbb{C}^d_{ic} \mathbf{G}_{db}$. Therefore, we can show that the following representation holds:

$$\mathrm{ad}^{\dagger}_{\xi} \xi = \sum_{i,b} \alpha_i \alpha_b \sum_a \left( \sum_{c,d} \mathbf{G}^{ac} \mathbb{C}^d_{ic} \mathbf{G}_{db} \right) \mathrm{E}_a.$$

Comparing coefficients in the geodesic equation $\sum_a \dot{\alpha}_a \mathrm{E}_a + \sum_a \left( \sum_{i,b} \alpha_i \alpha_b \mathbf{G}^{ac} \mathbb{C}^d_{ic} \mathbf{G}_{db} \right) \mathrm{E}_a = 0$ yields, for each $a$,

$$\dot{\alpha}_a = - \sum_{i,b,c,d} \alpha_i \alpha_b \mathbf{G}^{ac} \mathbb{C}^d_{ic} \mathbf{G}_{db}.$$

This is exactly the claimed coefficient ODE. Hence the parametrized field $\xi_\theta$ satisfies the geodesic equation if and only if the coefficients obey the stated system. $\qquad \square$

## B.2. Extended Lie Algebra and Exterior Algebra

**Definition B.5** (Generalized Parallel Transport Group). Let $G$ be a Lie group with Lie algebra $\mathfrak{g}$, and define the $R$ truncated exterior algebra of $\mathfrak{g}$ by $\mathfrak{G}^{\leq R}(\mathfrak{g}) := \mathfrak{g} \oplus (\mathfrak{g} \wedge \mathfrak{g}) \oplus (\mathfrak{g} \wedge \mathfrak{g} \wedge \mathfrak{g}) \oplus \cdots \oplus (\mathfrak{g}^{\wedge R})$. We then define the *generalized parallel transport group* as the semidirect product

$$\mathfrak{H}_{\mathrm{PT}}^{\leq R} := G \ltimes \mathfrak{G}^{\leq R}, \quad (g, Z) \quad \text{with} \quad Z := (Z^{(1)}, Z^{(2)}, \dots, Z^{(R)}), \quad Z^{(r)} \in \mathfrak{g}^{\wedge r} = \Lambda(V)^{\leq r}, \quad r \leq R$$

The binary operation on $\mathfrak{H}_{\mathrm{PT}}^{\leq R}$ is defined recursively by a graded semidirect product: for any two elements $(g, Z), (h, Y) \in \mathfrak{H}_{\mathrm{PT}}^{\leq R}$,

$$(g, Z) \ltimes (h, Y) = \big(gh, Z \star (g \curvearrowright Y)\big),$$

where the group action $g \curvearrowright Y$ and graded product $\star$ acts on the algebraic components are given by

$$g \curvearrowright Y := \big(\mathrm{Ad}_g Y^{(1)}, \mathrm{Ad}_g Y^{(2)}, \dots, \mathrm{Ad}_g Y^{(R)}\big), \quad (Z \star Y)^{(r)} = Z^{(r)} + Y^{(r)} + \frac{1}{r}\sum_{a=1}^{r-1} Z^{(a)} \wedge Y^{(r-a)}, \quad 2 \leq r \leq R.$$

We now provide an explicit generalization of the previously introduced parallel transport framework in Definition 4.1 originally defined for the simpler case $R = 2$ to an arbitrary truncation level $R$. Specifically, the generalized parallel transport object is defined as the tuple

$$\mathcal{S}_m := \left( \underbrace{\mathcal{G}_m, \, \mathcal{U}_m, \, \mathcal{C}_m}_{R=2, \text{ Definition } 4.1}, \mathcal{Z}_m^{(3)}, \dots, \mathcal{Z}_m^{(R)} \right) \in G \ltimes \mathfrak{g} \ltimes (\mathfrak{g} \wedge \mathfrak{g}) \ltimes \cdots \ltimes (\mathfrak{g}^{\wedge R}),$$

where $\mathcal{G}_m \in G$ is the cumulative group-level transformation, $\mathcal{U}_m \in \mathfrak{g}$ is the first-order velocity component, $\mathcal{C}_m \in \mathfrak{g} \wedge \mathfrak{g}$ is the second-order curvature component, $\mathcal{Z}_m^3 \in \mathfrak{g}^{\wedge 3}$ is the third-order volume component and so on up to degree $R$. For a small step $\Delta \mathcal{S}_m = (g_m, U_m, C_m, Z_m^{(3)}, \dots, Z_m^{(R)})$, the corresponding recursive update is defined via a graded semidirect product:

$$\textbf{(Parallel Prefix of Graded Algebra).} \qquad \mathcal{S}_{m+1} = \mathcal{S}_m \ltimes \Delta \mathcal{S}_m. \tag{24}$$

The update of each algebraic level in the generalized parallel transport framework follows a graded, adjoint-propagated accumulation rule. These rules explicitly capture how higher-order geometric and algebraic structures accumulate when tracking parallel transport on a Lie group. Formally, the discrete updates from step $m$ to $k+1$ are succinctly given by:

$$\mathcal{G}_{m+1} = \mathcal{G}_m g_m, \tag{25}$$

$$\mathcal{U}_{m+1} = \mathcal{U}_m + \mathrm{Ad}_{\mathcal{G}_m} \mathrm{U}_m, \tag{26}$$

$$\mathcal{C}_{m+1} = \mathcal{C}_m + \mathrm{Ad}_{\mathcal{G}_m} \mathrm{C}_m + \frac{1}{2}\mathcal{U}_m \wedge \mathrm{Ad}_{\mathcal{G}_m} \mathrm{U}_m, \tag{27}$$

$$\mathcal{Z}_{m+1}^{(3)} = \mathcal{T}_m + \mathrm{Ad}_{\mathcal{G}_m} Z_m^{(3)} + \frac{1}{3}\big(\mathcal{U}_m \wedge \mathrm{Ad}_{G_m} \mathrm{C}_m + \mathcal{C}_m \wedge \mathrm{Ad}_{\mathcal{G}_m} U_m\big), \tag{28}$$

$$\mathcal{Z}_{m+1}^{(4)} = \mathcal{Z}_m^{(4)} + \mathrm{Ad}_{\mathcal{G}_m} \mathrm{Z}_m^{(4)} + \frac{1}{4}\big(\mathcal{U}_m \wedge \mathrm{Ad}_{\mathcal{G}_m} \mathrm{Z}_m^{(3)} + \mathcal{C}_m \wedge \mathrm{Ad}_{G_m} \mathrm{C}_m + \mathcal{T}_m \wedge \mathrm{Ad}_{\mathcal{G}_m} \mathrm{U}_m\big). \tag{29}$$

Each line systematically encodes geometric accumulation at its corresponding algebraic order. The group update $\mathcal{G}_{m+1} = \mathcal{G}_m g_m$ aggregates total displacement on the Lie group, while the velocity update $\mathcal{U}_{m+1}$ advects the incoming step $U_m$ through the adjoint action. The curvature term $\mathcal{C}_{m+1}$ captures second–order noncommutativity between infinitesimal displacements, and the third–order term $\mathcal{Z}_{m+1}^{(3)}$ incorporates higher–order antisymmetric holonomy effects. These graded updates recursively encode multi–level geometric accumulation, generalizing naturally to an arbitrary algebraic order $r$.

$$\mathcal{Z}_{m+1}^{(r)} = \mathcal{Z}_m^{(r)} + \mathrm{Ad}_{G_m} Z_m^{(r)} + \frac{1}{r}\sum_{a=1}^{r-1} \mathcal{Z}_m^{(a)} \wedge \mathrm{Ad}_{G_m} Z_m^{(r-a)}, \qquad 2 \leq r \leq R. \tag{30}$$

Here, the wedge product captures intrinsic noncommutativity, and the adjoint operation $\mathrm{Ad}_{G_m}$ ensures global-frame alignment across the discrete steps since it preserves multi-linearity: $\mathrm{Ad}_{G_m}(X_1 \wedge \cdots \wedge X_r) = (\mathrm{Ad}_{G_m} X_1) \wedge \cdots \wedge (\mathrm{Ad}_{G_m} X_r)$.

**Local Coordinate Expression.** To further clarify this construction, we explicitly derive its representation in local coordinates. Let $\{e_i(x)\}_{i=1}^d$ be a local moving frame and write the adjoint action in this frame by $(A_m)_j^i = (\mathrm{Ad}_{G_m})_j^i$. For multi–indices $I = (i_1, \ldots, i_r)$, the $r$-th order term can be generalized as

$$\mathcal{Z}_m^{(r)} = \frac{1}{r!} \mathcal{Z}_{m,i_1 \cdots i_r}^{(r)} e_{i_1} \wedge \cdots \wedge e_{i_r}, \quad Z_k^{(r)} = \frac{1}{r!} Z_{m,i_1 \cdots i_r}^{(r)} e_{i_1} \wedge \cdots \wedge e_{i_r}. \tag{31}$$

From the construction, the induced representation on the exterior power $\wedge^r \mathfrak{g}$, the vector space consisting of fully antisymmetric $r$-linear combinations of elements from $\mathfrak{g}$, is explicitly given by

$$(\wedge^r A_m)_{i_1 \cdots i_r}{}^{j_1 \cdots j_r} = A_{m,[i_1}{}^{j_1} \cdots A_{m,i_r]}{}^{j_r} = \frac{1}{r!} \sum_{\sigma \in S_r} \mathrm{sgn}(\sigma) A_{m,i_{\sigma(1)}}{}^{j_1} \cdots A_{m,i_{\sigma(r)}}{}^{j_r},$$

where the square brackets $[\cdots]$ denote full antisymmetrization of the enclosed indices. Using this explicit antisymmetric representation, we generalize the recursive algebraic update rule to arbitrary higher-order terms. Specifically, for each integer order $r$ with $2 \leq r \leq R$, the $r$th-order algebraic accumulation $\mathcal{Z}_m^{(r)}$ in (31) can be rewritten component-wise as follows:

$$\mathcal{Z}_m^{(r)} = \left( \mathcal{Z}_m^{(r)} + \mathrm{Ad}_{\mathcal{G}_m} Z_m^{(r)} + \frac{1}{r} \sum_{a=1}^{r-1} \mathcal{Z}_m^{(a)} \wedge \mathrm{Ad}_{\mathcal{G}_m} Z_m^{(r-a)} \right)_{i_1 \cdots i_r}$$

$$= \mathcal{Z}_{k,i_1 \cdots i_r}^{(r)} + (\wedge^r A_m)_{i_1 \cdots i_r}{}^{j_1 \cdots j_r} Z_{k,j_1 \cdots j_r}^{(r)} + \frac{1}{r} \sum_{a=1}^{r-1} \frac{r!}{a!(r-a)!} \mathcal{Z}_{m,[i_1 \cdots i_a}^{(a)} (\wedge^{r-a} A_m)_{i_{a+1} \cdots i_r]}{}^{j_1 \cdots j_{r-a}} Z_{k,j_1 \cdots j_{r-a}}^{(r-a)},$$

where again the bracket notation $[\cdots]$ indicates full antisymmetrization over the enclosed indices. Now, we are ready to provide explicit description of parameterization of $\mathcal{Z}_m^{(r)}$ in local coordinate:

---

**Proposition B.6** (Parameterized Higher–Order Algebraic Accumulations). *Let $\xi_\theta(t) = \sum_{a=1}^{|\mathrm{Dim}(\mathbb{B})|} \alpha_a(t, z; \theta) E_a$ be a neural geodesic field as introduced in Proposition 3.1, inducing a left–trivialized flow map $\mathcal{G}_m = \exp(h\xi_\theta(t_m))$ with step size $h > 0$. Define the induced adjoint operator and its wedge representation by*

$$A_m(\theta) := \mathrm{Ad}_{\mathcal{G}_m}^\theta = \exp\left( h \, \mathrm{ad}_{\xi_\theta(t_m)} \right), \quad (\wedge^r A_m(\theta))_{i_1 \cdots i_r}{}^{j_1 \cdots j_r} = \frac{1}{r!} \sum_{\sigma \in S_r} \mathrm{sgn}(\sigma) A_{m,i_{\sigma(1)}}{}^{j_1}(\theta) \cdots A_{m,i_{\sigma(r)}}{}^{j_r}(\theta).$$

*Then, the $r$th–order geometric accumulation $\mathcal{Z}_m^{(r)}(\theta)$ defined on the truncated tensor algebra $\mathfrak{G}^{\leq R}(\mathfrak{g})$ is parameterized in terms of the neural geodesic coefficients $\alpha_a(t, z; \theta)$ as*

$$\mathcal{Z}_m^{(r)}(\theta) = \left( \mathcal{Z}_m^{(r)}(\theta) + \mathrm{Ad}_{\mathcal{G}_m}^\theta Z_m^{(r)} + \frac{1}{r} \sum_{a=1}^{r-1} \mathcal{Z}_m^{(a)} \wedge \mathrm{Ad}_{\mathcal{G}_m}^\theta Z_m^{(r-a)} \right)_{i_1 \cdots i_r}$$

$$= \mathcal{Z}_{m,i_1 \cdots i_r}^{(r)}(\theta) + (\wedge^r A_m(\theta))_{i_1 \cdots i_r}{}^{j_1 \cdots j_r} Z_{m,j_1 \cdots j_r}^{(r)}(\theta)$$

$$+ \frac{1}{r} \sum_{a=1}^{r-1} \frac{r!}{a!(r-a)!} \mathcal{Z}_{m,[i_1 \cdots i_a}^{(a)} (\wedge^{r-a} A_m(\theta))_{i_{a+1} \cdots i_r]}{}^{j_1 \cdots j_{r-a}}(\theta) Z_{m,j_1 \cdots j_{r-a}}^{(r-a)}(\theta),$$

*where each adjoint block $A_m = \exp(h \, \mathrm{ad}_{\xi_\theta(t_m)})$ depends smoothly on the neural coefficients $\alpha_a(t_m, z; \theta)$ through the Lie algebraic structure constants $\mathbb{C}_{ic}^d$.*

---

**Error Analysis of Connection-Correction.** Building upon the mathematical framework established above, we now rigorously quantify the discretization error, assessing how closely our discrete parallel transport approximates the continuous exact parallel transport. We first formally define discretization of exact continuous parallel transport in local coordinate:

---

**Lemma B.7** (Discrete Parallel Transport). *Consider a discrete transport step from $m$ to $m+1$ on a Lie group $G$, where the Levi-Civita connection $\Gamma^i_{jk}(g_m)$ is defined at a group element $g_m \in G$. Let the infinitesimal displacement be characterized by the left–trivialized velocity components $\xi^k_m$. Then, the corresponding discrete parallel transport operator $\mathcal{PT}^\rho_{m\to m+1}$ acting on the local frame is approximated to first order as*

$$\left(\mathcal{PT}^\rho_{m\to m+1}\right)^i_j = \delta^i_j - h\Gamma^i{}_{kj}(g_m)\xi^k_m + O(h^2), \tag{32}$$

*where $h$ denotes the step size and higher-order connection corrections are absorbed into the $O(h^2)$ term.*

---

*Proof.* We begin from the definition of parallel transport of a vector field along a curve on a Riemannian manifold (or a Lie group with Levi–Civita connection). Let $v(t) \in T_{g(t)}G$ be a vector field transported along a smooth curve $g(t)$, satisfying the covariant derivative condition $\nabla_{\dot{g}(t)}v(t) = 0$. In local coordinates, this condition expands to the first–order ODE

$$\frac{dv^i(t)}{dt} + \Gamma^i{}_{kj}(g(t))\dot{g}^k(t)v^j(t) = 0.$$

Let the curve be discretized as $g_{m+1} = g_m \exp(h\xi_m)$, where $\xi_m \in \mathfrak{g}$ is the left–trivialized velocity and $h$ is the step size. Over a short interval $[t_m, t_{m+1}]$ of length $h$, we approximate $g(t) \approx g_m$ and $\dot{g}^k(t) \approx \xi^k_m$. The parallel–transport equation then simplifies to

$$\frac{dv^i}{dt} = -\Gamma^i{}_{kj}(g_m)\xi^k_m v^j + O(h) \quad \longrightarrow \quad v^i(t_{m+1}) = \left(\delta^i_j - h\Gamma^i{}_{kj}(g_m)\xi^k_m + O(h^2)\right)v^j(t_m),$$

where the result can be obtained by integrating linear ODE from $t_m$ to $t_{m+1} = t_m + h$. Thus, the parallel transport operator $\mathcal{PT}^\rho_{m\to m+1}: T_{g_m}G \to T_{g_{m+1}}G$ satisfies

$$(\mathcal{PT}^\rho_{m\to m+1})^i_j = \delta^i_j - h\Gamma^i{}_{kj}(g_m)\xi^k_m + O(h^2),$$

which corresponds exactly to the claimed expansion (32). The $O(h^2)$ term accounts for the neglected curvature and connection-variation contributions arising from the second–order Taylor expansion of the solution. $\qquad\square$

Equipped with Lemma B.7, we are now ready to introduce a mathematical object that precisely quantifies the geometric distortion between the discrete and exact continuous parallel transport, defined explicitly through a connection–corrected adjoint action.

---

**Definition B.8** (Connection–corrected Adjoint). Let $\mathcal{PT}^\rho_{m\to m+1}$ be a discrete parallel transport operator defined in Lemma B.7. Then, we define *connection–corrected adjoint action* as follows:

$$\mathrm{Ad}^\nabla_{\mathcal{G}_m} = \left(\mathcal{PT}^\rho_{m\to m+1}\right)^{-1}\mathrm{Ad}_{\mathcal{G}_m}\left(\mathcal{PT}^\rho_{m\to m+1}\right).$$

Its wedge representation admits the first–order local coordinate expansion

$$\left(\wedge^r \mathrm{Ad}^\nabla_{G_m}\right)^{j_1\cdots j_r}_{i_1\cdots i_r} = \left(\wedge^r \mathrm{Ad}_{G_m}\right)^{j_1\cdots j_r}_{i_1\cdots i_r} + h\sum_{s=1}^r \left(\mathrm{Ad}_{G_m}\right)^{j_1}_{i_1}\cdots\left([\mathrm{Ad}_{G_m},\Omega_m]\right)^{j_s}_{i_s}\cdots\left(\mathrm{Ad}_{G_m}\right)^{j_r}_{i_r} + O(h^2),$$

where the connection one-form $\Omega$ and the Lie bracket $[\cdot,\cdot]$ are given by $\Omega^i_{mj} = \Gamma^i{}_{kj}(g_m)\xi^k_m$ and $[\mathrm{Ad}_{G_m},\Omega_m] = \mathrm{Ad}_{G_m}\Omega_m - \Omega_m\mathrm{Ad}_{G_m}$.

---

**Lemma B.9** (Curvature–controlled bound on the connection one–form). *For any geodesic curve $g(t) \in G$ with left–trivialized velocity $\xi(t) \in \mathfrak{g}$, the corresponding connection one–form $\Omega^i_{mj} = \Gamma^i{}_{kj}(g_m)\xi^k_m$ satisfies the curvature–controlled bound $\|\Omega_m\| \leq C_G\sqrt{\kappa_{\max}}\|\xi_m\|$, where $C_G > 0$ depends only on the Lie group metric structure.*

*Proof.* Recall that for a Riemannian metric $\langle \cdot, \cdot \rangle_g$ on $\mathfrak{g}$, the sectional curvature for the plane spanned by $X, Y \in \mathfrak{g}$ is expressed as

$$K(X,Y) = \frac{\langle R(X,Y)Y, X \rangle_g}{\|X\|^2\|Y\|^2 - \langle X, Y \rangle_g^2} = \frac{1}{4}\|[X,Y]\|^2 - \frac{3}{4}\|\operatorname{sym}(\operatorname{ad}_X Y)\|^2,$$

where $\operatorname{sym}(A) = \frac{1}{2}(A + A^\dagger)$ denotes the symmetric part. Thus, if $|K(X,Y)| \leq \kappa_{\max}$ for all orthonormal $X, Y$, then we have $\|\operatorname{ad}_X\| \leq 2\sqrt{\kappa_{\max}}$. Next, we apply this to the local velocity direction $X = \xi_m$ to obtain

$$\|\Gamma(g_m)[\xi_m]\| = \tfrac{1}{2}\|\operatorname{ad}_{\xi_m} - \operatorname{ad}^\dagger_{\xi_m}\| \leq C_G\sqrt{\kappa_{\max}}\|\xi_m\|,$$

where the constant $C_G$ depends only on the specific normalization of the structure constants induced by the chosen metric. For a bi–invariant metric, $\operatorname{ad}^\dagger_{\xi_m} = -\operatorname{ad}_{\xi_m}$, giving $C_G = \frac{1}{2}$, while for a general left–invariant metric $C_G$ depends on the relative norm of $\operatorname{sym}(\operatorname{ad}_{\xi_m})$. Finally, since $\Omega_m = \Gamma(g_m)[\xi_m]$, the same bound holds for the connection one–form itself:

$$\|\Omega_m\| \leq C_G\sqrt{\kappa_{\max}}\|\xi_m\|.$$

$\square$

**Operator Norm.** Throughout the appendix, we aim to measure the local transport discrepancy using the operator norm induced by the Riemannian or Euclidean inner product on the representation space $V \simeq \mathbb{R}^d$. For any linear operator $A : V \to V$ with matrix components $A^i{}_j$, the operator norm is defined by

$$\|A\|_{\operatorname{op}} := \sup_{v \in V \setminus \{0\}} \frac{\|Av\|}{\|v\|} = \sqrt{\lambda_{\max}(A^\top A)},$$

where $\lambda_{\max}$ denotes the largest eigenvalue. Equivalently, if $V$ is equipped with the inner product $\langle \cdot, \cdot \rangle_V$, then

$$\|A\|_{\operatorname{op}} = \sup_{\|v\|_V = 1} \sqrt{\langle Av, Av \rangle_V}.$$

When comparing transport operators (*e.g.*, discrete, exact) $\mathcal{PT} : V \to V$, we thus measure their deviation by using

$$\|\mathcal{PT}^\rho_{m \to m+1} - \mathcal{PT}^\rho_{s \to t}\|_{\operatorname{op}} = \sup_{\|v\|_V = 1} \|(\mathcal{PT}^\rho_{m \to m+1} - \mathcal{PT}^\rho_{s \to t})v\|_V, \tag{33}$$

which quantifies the worst-case amplification factor of the difference acting on unit vectors. In the following theoretical finding, our goal is to show the upper-bound of (33):

**Proposition B.10** (Local Discrepancy under Connection–Corrected Adjoint). *Let us denote $[t_s, t_t] = [t_m, t_{m+1}]$ and $h = t_{m+1} - t_m$. For a sufficiently small step size $h$ and compact Lie group $G$, the local transport error measured in any submultiplicative operator norm $\|\cdot\|_{\operatorname{op}}$ satisfies*

$$\|\mathcal{PT}^\rho_{m \to m+1} - \mathcal{PT}^\rho_{s \to t}\|_{\operatorname{op}} \lesssim \mathcal{O}\left(h^2\left(\|\dot{\xi}_m\| + \frac{\operatorname{Diam}(G)^2}{h^2}\right)\right) + \mathcal{O}\left(h^2\sqrt{\kappa_{\max}}\operatorname{Diam}(G)\right),$$

*where $(\kappa_{\max}, \operatorname{Diam})$ denotes where $\kappa_{\max}$ denotes the maximum sectional curvature of $G$, and $\operatorname{Diam}(G)$ denotes the diameter of $G$ under a Riemannian metric.*

*Proof.* We begin with the connection–corrected adjoint defined by

$$\operatorname{Ad}^\nabla_{\mathcal{G}_m} = \left(\mathcal{PT}^\rho_{m \to m+1}\right)^{-1}\operatorname{Ad}_{\mathcal{G}_m}\left(\mathcal{PT}^\rho_{m \to m+1}\right), \tag{34}$$

where $\mathcal{PT}^\rho_{m\to m+1}$ is the discrete parallel transport operator described in Lemma B.7. The discrete transport and its inverse are expanded as $\mathcal{PT}^\rho_{m\to m+1} = I - h\Omega_m + O(h^2)$ and $\left(\mathcal{PT}^\rho_{m\to m+1}\right)^{-1} = I + h\Omega_m + O(h^2)$. Substituting these expansions into the definition (34) gives

$$
\begin{aligned}
\mathrm{Ad}^\nabla_{\mathcal{G}_m} &= e^{h\Omega_m}\,\mathrm{Ad}_{\mathcal{G}_m}\,e^{-h\Omega_m} \\
&= \Big(\sum_{r=0}^\infty \frac{h^r}{r!}\,\mathrm{ad}^r_{\Omega_m}\Big)(\mathrm{Ad}_{\mathcal{G}_m}) \\
&= \mathrm{Ad}_{\mathcal{G}_m} + h[\Omega_m, \mathrm{Ad}_{\mathcal{G}_m}] + \frac{h^2}{2!}[\Omega_m, [\Omega_m, \mathrm{Ad}_{\mathcal{G}_m}]] + \frac{h^3}{3!}[\Omega_m, [\Omega_m, [\Omega_m, \mathrm{Ad}_{\mathcal{G}_m}]]] + \cdots \\
&= \mathrm{Ad}_{\mathcal{G}_m} + \sum_{r=1}^\infty \frac{h^r}{r!}\,\mathrm{ad}^r_{\Omega_m}(\mathrm{Ad}_{\mathcal{G}_m}),
\end{aligned}
\tag{35}
$$

where $\mathrm{ad}^r_{\Omega_m}(\mathrm{Ad}_{\mathcal{G}_m}) = [\Omega_m, [\Omega_m, \ldots, [\Omega_m, \mathrm{Ad}_{\mathcal{G}_m}]\ldots]]$ denotes the $r$-fold nested commutator. Let $\mathcal{PT}^\rho_{s\to t}$ denote the exact continuous parallel transport over the same time interval, generated by the continuous geodesic velocity $\xi(t)$. The continuous evolution admits the series representation

$$
\mathcal{PT}^\rho_{s\to t} = \exp\left(-h\rho_*(\mathrm{ad}^\dagger_{\xi_m}) - \frac{h^2}{2}\rho_*\big(\dot\xi_m + [\xi_m, \mathrm{ad}^\dagger_{\xi_m}]\big) + \mathcal{O}(h^3)\right),
$$

$$
\mathcal{PT}^\rho_{m\to m+1} = \exp\big(-h\rho_*(\mathrm{ad}^\dagger_{\xi_m})\big) + O(h^2).
$$

where $\xi_m = \xi(t_m)$ denotes the left–trivialized velocity of the geodesic. Now, we apply the standard operator inequality $\|e^A - e^B\| \le e^{\max(\|A\|, \|B\|)}\|A - B\|$ to the two exponential parallel transport operators. Hence, for the $A_m = h\rho_*\big(\mathrm{ad}^\dagger_{\xi_m}\big)$ and $B_m = h\rho_*\big(\mathrm{ad}^\dagger_{\xi_m}\big) + \frac{h^2}{2}\rho_*\big(\dot\xi_m + [\xi_m, \mathrm{ad}^\dagger_{\xi_m}]\big) + \mathcal{O}(h^3)$, we have

$$
\big\|\mathcal{PT}^\rho_{m\to m+1} - \mathcal{PT}^\rho_{s\to t}\big\|_{\mathrm{op}} = \big\|\exp(-A_m) - \exp(-B_m)\big\|_{\mathrm{op}}
\tag{36}
$$

$$
\le e^{\max(\|A_m\|_{\mathrm{op}}, \|B_m\|_{\mathrm{op}})}\|A_m - B_m\|_{\mathrm{op}}
\tag{37}
$$

$$
\le e^{\max(\|A_m\|_{\mathrm{op}}, \|B_m\|_{\mathrm{op}})}\left(\frac{h^2}{2}\|\rho_*\|_{\mathrm{op}}\|\dot\xi_m + [\xi_m, \mathrm{ad}^\dagger_{\xi_m}]\|_{\mathrm{op}} + h^2\|\rho_*\|_{\mathrm{op}}\|\mathrm{Ad}^\nabla_{\mathcal{G}_m} - \mathrm{Ad}_{\mathcal{G}_m}\|_{\mathrm{op}}\right) + \mathcal{O}(h^3)
\tag{38}
$$

$$
\le e^{\max(\|A_m\|_{\mathrm{op}}, \|B_m\|_{\mathrm{op}})}\left(\frac{h^2}{2}\|\rho_*\|_{\mathrm{op}}\|\dot\xi_m + [\xi_m, \mathrm{ad}^\dagger_{\xi_m}]\|_{\mathrm{op}} + h^2\|\rho_*\|_{\mathrm{op}}\big(e^{2h\|\Omega_m\|} - 1\big)\|\mathrm{Ad}_{\mathcal{G}_m}\|_{\mathrm{op}}\right) + \mathcal{O}(h^3)
\tag{39}
$$

$$
\le e^{h\|\rho_*\|_{\mathrm{op}}\|\mathrm{ad}^\dagger_{\xi_m}\|_{\mathrm{op}}}\left(\frac{h^2}{2}\|\rho_*\|_{\mathrm{op}}\|\dot\xi_m + [\xi_m, \mathrm{ad}^\dagger_{\xi_m}]\|_{\mathrm{op}} + h^2\|\rho_*\|_{\mathrm{op}}\big(e^{2hC_G\sqrt{\kappa_{\max}}\|\xi_m\|} - 1\big)\|\mathrm{Ad}_{\mathcal{G}_m}\|_{\mathrm{op}}\right) + \mathcal{O}(h^3)
\tag{40}
$$

$$
\le e^{h\|\rho_*\|\|\mathrm{ad}^\dagger_{\xi_m}\|}\left[\frac{h^2}{2}\|\rho_*\|\|\dot\xi_m + [\xi_m, \mathrm{ad}^\dagger_{\xi_m}]\| + 2h^3 C_G\sqrt{\kappa_{\max}}\|\rho_*\|\|\xi_m\|\|\mathrm{Ad}_{\mathcal{G}_m}\|\right] + \mathcal{O}(h^3).
\tag{41}
$$

By taking the operator norm and applying the previously established relation in (35), the connection one–form $\Omega_m$ satisfies the curvature–controlled bound $\|\Omega_m\| \le C_G\sqrt{\kappa_{\max}}\|\xi_m\|$ in Lemma B.9 which gives

$$
\|\mathrm{Ad}^\nabla_{\mathcal{G}_m} - \mathrm{Ad}_{\mathcal{G}_m}\|_{\mathrm{op}} \le \sum_{r=1}^\infty \frac{h^r}{r!}\|\mathrm{ad}^r_{\Omega_m}(\mathrm{Ad}_{\mathcal{G}_m})\|_{\mathrm{op}} \le \big(e^{2h\|\Omega_m\|} - 1\big)\|\mathrm{Ad}_{\mathcal{G}_m}\|_{\mathrm{op}} \le \left(e^{2hC_G\sqrt{\kappa_{\max}}\|\xi_m\|} - 1\right)\|\mathrm{Ad}_{\mathcal{G}_m}\|_{\mathrm{op}},
$$

where we used the fact that $\|\mathrm{ad}^r_{\Omega_m}(X)\| \le (2\|\Omega_m\|)^r\|X\|$. This relation shows the third and fourth inequalities in (39) and (40).

$$
\big\|\mathcal{PT}^\rho_{m\to m+1} - \mathcal{PT}^\rho_{s\to t}\big\|_{\mathrm{op}} \le e^{h\|\rho_*\|\|\mathrm{ad}^\dagger_{\xi_m}\|}\left[\frac{h^2}{2}\|\rho_*\|\|\dot\xi_m + [\xi_m, \mathrm{ad}^\dagger_{\xi_m}]\| + 2h^3 C_G\sqrt{\kappa_{\max}}\|\rho_*\|\|\xi_m\|\|\mathrm{Ad}_{\mathcal{G}_m}\|\right] + \mathcal{O}(h^3)
$$

$$
\le h^2\|\rho_*\|e^{h\|\rho_*\|\|\mathrm{ad}^\dagger_{\xi_m}\|}\left[\frac{1}{2}\|\dot\xi_m + [\xi_m, \mathrm{ad}^\dagger_{\xi_m}]\| + 2hC_G\sqrt{\kappa_{\max}}\|\xi_m\|\|\mathrm{Ad}_{\mathcal{G}_m}\|\right] + \mathcal{O}(h^3)
$$

$$
\lesssim h^2\|\rho_*\|e^{h\|\rho_*\|\|\mathrm{ad}^\dagger_{\xi_m}\|}\big(\|\dot\xi_m\| + \|\xi_m\|^2 + h\sqrt{\kappa_{\max}}\|\xi_m\|\big) + \mathcal{O}(h^3).
$$

Summarizing the obtained result above, we have

$$\left\|\mathcal{PT}^{\rho}_{m\to m+1} - \mathcal{PT}^{\rho}_{s\to t}\right\|_{\mathrm{op}} \lesssim \mathcal{O}\Big(h^2\|\rho_*\|(\|\dot{\xi}_m\| + \|\xi_m\|^2)\Big) + \mathcal{O}\Big(h^3\sqrt{\kappa_{\max}}\|\rho_*\|\|\xi_m\|\Big).$$
$$\le \mathcal{O}\Big(h^2(\|\dot{\xi}_m\| + \tfrac{\ell_{\max}^2}{h^2})\Big) + \mathcal{O}\Big(h^2\sqrt{\kappa_{\max}}\ell_{\max}\Big), \tag{42}$$

where the second equality follows as the differential $\rho_* : \mathfrak{g} \to \mathfrak{m}(V)$ acts as an isometry, yielding $\|\rho_*\| \le 1$, and since the discrete update follows $g_{m+1} = g_m \exp(h\xi_m)$, the geodesic distance satisfies $\mathrm{dist}(g_m, g_{m+1}) = h\|\xi_m\| + O(h^2)$. By letting $\ell_m := \mathrm{dist}(g_m, g_{m+1})$ and assuming a uniform upper bound $\ell_m \le \ell_{\max}$ gives $\|\xi_m\| \le \frac{\ell_{\max}}{h} \le \frac{\mathrm{Diam}(G)}{h}$. $\qquad\square$

---

**Corollary B.11** (Global Cumulative Transport Bound). *After $m + 1$ successive discrete steps, the cumulative transport error between the discrete and continuous parallel transports from $0$ to $m + 1$ satisfies*

$$\left\|\mathcal{PT}^{\rho}_{0\to m+1} - \mathcal{PT}^{\rho}_{0\to t_{m+1}}\right\|_{\mathrm{op}} \lesssim (m+1)\Big[\mathcal{O}\Big(h^2(\|\dot{\xi}\|_{\max} + \tfrac{\mathrm{Diam}(G)^2}{h^2})\Big) + \mathcal{O}\Big(h^2\sqrt{\kappa_{\max}}\mathrm{Diam}(G)\Big)\Big],$$

*where $\|\dot{\xi}\|_{\max} := \max_{0\le k\le m} \|\dot{\xi}_k\|$.*

---

*Proof.* The total discrete parallel transport from step $0$ to $m + 1$ is given by the composition

$$\mathcal{PT}^{\rho}_{0\to m+1} = \mathcal{PT}^{\rho}_{m\to m+1}\mathcal{PT}^{\rho}_{m-1\to m}\cdots\mathcal{PT}^{\rho}_{0\to 1},$$

and similarly for the continuous transport $\mathcal{PT}^{\rho}_{0\to t_{m+1}}$. Using the submultiplicative property of the operator norm and applying a telescoping expansion, one obtains

$$\left\|\mathcal{PT}^{\rho}_{0\to m+1} - \mathcal{PT}^{\rho}_{0\to t_{m+1}}\right\|_{\mathrm{op}} \le \sum_{k=0}^{m} \left\|\mathcal{PT}^{\rho}_{k\to k+1} - \mathcal{PT}^{\rho}_{s_k\to t_k}\right\|_{\mathrm{op}} \prod_{j\ne k}\|\mathcal{PT}^{\rho}_{j\to j+1}\|_{\mathrm{op}} + \mathcal{O}(h^3).$$

Since each $\|\mathcal{PT}^{\rho}_{j\to j+1}\|_{\mathrm{op}} = 1 + \mathcal{O}(h)$ by construction of the parallel transport, we obtain

$$\left\|\mathcal{PT}^{\rho}_{0\to m+1} - \mathcal{PT}^{\rho}_{0\to t_{m+1}}\right\|_{\mathrm{op}} \lesssim \sum_{k=0}^{m} \left\|\mathcal{PT}^{\rho}_{k\to k+1} - \mathcal{PT}^{\rho}_{s_k\to t_k}\right\|_{\mathrm{op}} + \mathcal{O}(h^3).$$

Applying Proposition B.10 to each step $k$ yields

$$\left\|\mathcal{PT}^{\rho}_{k\to k+1} - \mathcal{PT}^{\rho}_{s_k\to t_k}\right\|_{\mathrm{op}} \lesssim \mathcal{O}\Big(h^2(\|\dot{\xi}_k\| + \tfrac{\mathrm{Diam}(G)^2}{h^2})\Big) + \mathcal{O}\Big(h^2\sqrt{\kappa_{\max}}\mathrm{Diam}(G)\Big).$$

Summing over $k = 0, \ldots, m$ and assuming bounded local variation $\|\dot{\xi}_k\| \le \|\dot{\xi}\|_{\max}$ gives

$$\left\|\mathcal{PT}^{\rho}_{0\to m+1} - \mathcal{PT}^{\rho}_{0\to t_{m+1}}\right\|_{\mathrm{op}} \lesssim (m+1)\Big[\mathcal{O}\Big(h^2(\|\dot{\xi}\|_{\max} + \tfrac{\mathrm{Diam}(G)^2}{h^2})\Big) + \mathcal{O}\Big(h^2\sqrt{\kappa_{\max}}\mathrm{Diam}(G)\Big)\Big],$$

which establishes the global cumulative upper bound. $\qquad\square$

## C. Discrete Parallel Transport with Parallel Scan

**Parallel Prefix on a Graded Semidirect Monoid.** Let $(\mathcal{H}, \odot, \mathrm{Id})$ denote an associative algebraic system equipped with an identity element. In our specific context, we focus on the graded semidirect product structure $\mathcal{H} = \mathfrak{H}_{\mathrm{PT}}^{\leq R}$, where the binary operation $\odot$ is defined by the recursive graded semidirect update. Given an ordered sequence of increments $\{\Delta \mathsf{S}_k\}_{k=1}^M \subset \mathcal{H}$, the inclusive prefixes are naturally defined as

$$\mathsf{P}_m := \Delta \mathsf{S}_1 \odot \Delta \mathsf{S}_2 \odot \cdots \odot \Delta \mathsf{S}_m, \quad m = 1, \ldots, M.$$

To efficiently compute all prefixes $\{\mathsf{P}_m\}_{m=1}^M$ with logarithmic parallel depth $O(\log M)$, we utilize a balanced binary-tree parallel prefix-scan algorithm, comprising an *upsweep* and a *downsweep* stage.

**Upsweep (Balanced Tree Reduction).** We initiate the upsweep by placing the original increments as the leaves at level zero, and recursively combine pairs of elements at each higher level, forming intermediate composites that represent progressively longer sub-block prefixes:

$$\mathsf{B}_k^{(0)} := \Delta \mathsf{S}_k, \qquad \mathsf{B}_j^{(\ell)} := \mathsf{B}_{2j-1}^{(\ell-1)} \odot \mathsf{B}_{2j}^{(\ell-1)}, \quad \ell = 1, \ldots, L, j = 1, \ldots, \left\lceil \frac{M}{2^\ell} \right\rceil,$$

where $L := \lceil \log_2 M \rceil$. Upon completion of the upsweep, the root node $\mathsf{B}_1^{(L)}$ accumulates the total product:

$$\mathsf{B}_1^{(L)} = \Delta \mathsf{S}_1 \odot \Delta \mathsf{S}_2 \odot \cdots \odot \Delta \mathsf{S}_M.$$

**Downsweep (Prefix Distribution).** In the downsweep stage, we distribute accumulated prefixes downwards from the root to leaves. Starting from the neutral identity element $\mathrm{Id}$ at the root, prefixes are recursively propagated to lower levels:

$$\mathsf{D}_1^{(L)} := \mathrm{Id}, \quad \mathsf{D}_{2j-1}^{(\ell-1)} := \mathsf{D}_j^{(\ell)}, \quad \mathsf{D}_{2j}^{(\ell-1)} := \mathsf{D}_j^{(\ell)} \odot \mathsf{B}_{2j-1}^{(\ell-1)}, \quad \ell = L, \ldots, 1, j = 1, \ldots, \left\lceil \frac{M}{2^\ell} \right\rceil.$$

At the leaves (level 0), we obtain all inclusive prefixes simultaneously and in parallel as:

$$\mathsf{P}_k = \mathsf{D}_k^{(0)} \odot \mathsf{B}_k^{(0)} = \mathsf{D}_k^{(0)} \odot \Delta \mathsf{S}_k, \quad k = 1, \ldots, M.$$

**Specialization to Euler–Poincaré Transport.** In our Euler–Poincaré framework, the general construction specializes naturally. We set the algebraic structure as the generalized parallel transport group, specifically $\mathcal{H} = \mathfrak{H}_{\mathrm{PT}}^{\leq R}$. Each increment is thus expressed explicitly as a tuple

$$\Delta \mathsf{S}_k = (g_k, Z_k^{(1)}, Z_k^{(2)}, \ldots, Z_k^{(R)}) \in G \ltimes \mathfrak{G}^{\leq R}.$$

Here, the binary operation $\odot$ corresponds exactly to the graded semidirect update defined in our geometric setting, and the identity element $\mathrm{Id}$ is explicitly given by the neutral tuple $(e, 0, \ldots, 0)$. Specifically, the parallel prefix-scan returns cumulative geometric states as the inclusive prefixes:

$$\mathsf{P}_m = \mathcal{S}_m = (G_m, \mathcal{U}_m, \mathcal{C}_m, \ldots, \mathcal{Z}_m^{(R)}), \quad m = 1, \ldots, M.$$

This construction not only maintains rigorous geometric consistency at every step and algebraic grade, but also achieves substantial computational acceleration by exploiting the inherent associativity of the underlying semidirect product structure, resulting in parallel computational complexity of $O(\log M)$ depth and $O(M)$ total operations. Such a formulation thereby significantly reduces the computational cost of geometric integrations in long-horizon simulations while naturally preserving fundamental physical invariants encoded by the Euler–Poincaré structure.

## D. Experimental Details

**Data generation.** Molecular trajectories are generated for two Villin variants, TRP-CAGE, and a $Mg^{2+}$ hydration system using OpenMM with the Amber14SB all-atom force field Maier et al. (2015) and TIP3P-FB water, with initial velocities drawn from the Maxwell-Boltzmann distribution for all systems Leimkuhler & Matthews (2013). All peptide systems use a $1.0$ nm solvent shell and a Langevin integrator with a $2$ fs time step, with coordinates saved every $0.1$ ps for a total of $10^5$ frames. The first Villin variant is constructed from a synthetic sequence-based model generated by `PeptideBuilder` with missing atoms completed by `PDBFixer` and simulated at $300$ K, while the second uses the experimentally resolved structure from PDB 1YRI RCSB Protein Data Bank (a) and is simulated at $600$ K. TRP-CAGE is initialized from PDB 2JOF RCSB Protein Data Bank (b) and simulated at $600$ K under the same protocol. The $Mg^{2+}$ hydration system uses ion parameters optimized for divalent cations Mamatkulov & Schwierz (2018) in a periodic cubic box of side length $4.0$ nm, propagated for $10$ ns at $300$ K with a $1$ fs time step and sampled every $0.1$ ps.

**Model architecture.** Each configuration $x_t \in \mathbb{R}^{3N}$ is encoded into a latent vector $z_t \in \mathbb{R}^{d_{\mathrm{lat}}}$ by a two-layer MLP and a symmetric two-layer MLP decoder maps latent states back to coordinate space while temporal generator coefficients $\alpha(t) \in \mathbb{R}^r$ are produced by a sinusoidal time embedding with $n_{\mathrm{freq}}$ frequencies followed by a two-layer MLP giving per-step Lie algebra increments $\Delta U_m = h\,\alpha(t_m)$ for $h = T/M$. We fix a basis $\{g_a\}_{a=1}^r$ of the Lie algebra $\mathfrak{g}$ and work on the truncated exterior algebra $\mathfrak{G}^{\leq R}(\mathfrak{g}) = \mathfrak{g} \oplus (\mathfrak{g} \wedge \mathfrak{g}) \oplus \cdots \oplus (\mathfrak{g}^{\wedge R})$ so that the parallel-transport state at step $m$ is $\mathcal{S}_m = \left(\mathcal{G}_m, \mathcal{U}_m, \mathcal{C}_m, \mathcal{Z}_m^{(3)}, \dots, \mathcal{Z}_m^{(R)}\right) \in G \ltimes \mathfrak{G}^{\leq R}(\mathfrak{g})$, with $\mathcal{U}_m \in \mathfrak{g}$, $\mathcal{C}_m \in \mathfrak{g} \wedge \mathfrak{g}$ and $\mathcal{Z}_m^{(r)} \in \mathfrak{g}^{\wedge r}$ for $3 \leq r \leq R$. The Lie algebra structure constants $C_{ij}^k$ in this basis are computed once by inner-product projection and are held fixed during training and induce the higher-order exterior powers and wedge interactions used in the update rules for $\mathfrak{G}^{\leq R}(\mathfrak{g})$. The latent dynamics follow a graded semidirect-product update $\mathcal{S}_{m+1} = \mathcal{S}_m \ltimes \Delta \mathcal{S}_m$ where each algebraic level is updated by $\mathcal{Z}_{m+1}^{(r)} = \mathcal{Z}_m^{(r)} + \mathrm{Ad}_{\mathcal{G}_m} Z_m^{(r)} + \frac{1}{r}\sum_{a=1}^{r-1} \mathcal{Z}_m^{(a)} \wedge \mathrm{Ad}_{\mathcal{G}_m} Z_m^{(r-a)}$, $2 \leq r \leq R$, and all cumulative generator coordinates are computed in parallel through an associative prefix-scan on $\mathfrak{H}_{\mathrm{PT}}^{\leq R}$. Latent states are updated by applying a truncated Lie-group exponential $z_{m+1} = \exp_{\mathcal{G}}\left(\mathcal{U}_m + \Phi(\mathcal{C}_m, \mathcal{Z}_m^{(3)}, \dots, \mathcal{Z}_m^{(R)})\right) z_m$, approximated by a fixed-order series expansion and a global inverse trajectory is obtained by composing the cumulative generators in reverse direction.

**Training setup.** All experiments are conducted on a single RTX 5000 Ada GPU using Python 3.10.18 with PyTorch 2.5.1 and CUDA 12.4, and we enable CUDA synchronization during evaluation to ensure accurate timing. We train all models with the Adam optimizer using a learning rate of $10^{-3}$ and default momentum parameters without learning rate decay, and gradients are clipped to unit $\ell_2$ norm with all computations performed in single-precision floating point. Table 3 summarizes the architectural and training hyperparameters used across datasets.

*Table 3.* Hyperparameters used for EPND across datasets.

| Dataset | $M$ | $d_{\mathrm{lat}}$ | $r$ | Hidden | $n_{\mathrm{freq}}$ | Series | $\alpha_{\mathrm{scale}}$ | Iters |
|---|---|---|---|---|---|---|---|---|
| Villin (100k) | 100,000 | 32 | 4 | 250 | 8 | 12 | 0.1 | 30,000 |
| Villin (1k) | 1,000 | 32 | 4 | 250 | 8 | 6 | 0.1 | 3,000 |
| TRP-CAGE (100k) | 100,000 | 32 | 4 | 250 | 8 | 12 | 0.1 | 100,000 |
| TRP-CAGE (1k) | 1,000 | 32 | 4 | 250 | 8 | 6 | 0.1 | 3,000 |
| Hydration (100k) | 100,000 | 32 | 8 | 250 | 8 | 12 | 0.1 | 100,000 |
| Hydration (1k) | 1,000 | 32 | 4 | 250 | 8 | 6 | 0.1 | 3,000 |

**Baseline implementations.** All baseline methods follow the authors' official public implementations and we preserve the published architectures, loss functions and optimization procedures whenever they remain compatible with molecular coordinate data. To ensure a standardized evaluation every baseline is adapted to operate on the same preprocessed numpy trajectories used for EPND by applying the identical $(T, N, 3) \to \mathbb{R}^{3N}$ flattening procedure and identical channelwise normalization statistics and methods that require momentum information receive $(T, N, 6)$ inputs constructed with the same normalization for both coordinates and momenta so that any performance difference reflects model behavior rather than inconsistencies in data handling. All temporal hyperparameters including the rollout length $M$ and the resulting training window are matched exactly across models and for baselines originally designed for shorter sequences only horizon related quantities such as prediction length or integration steps are modified while architectural components remain unchanged. When a model exhibit numerical instability on long molecular trajectories we adjust only scalar optimization parameters that improve stability such as lowering the learning rate or increasing the number of training iterations and no additional regularization or architectural alterations are introduced. Several baselines specify hidden widths or layer sizes that differ

substantially from those used by EPND and in such cases we adjust only the numerical widths of layers already present in the published architecture so that representational capacity becomes comparable without modifying the model structure.

- **KoNODE**: https://github.com/Baitie00/KoNODE
- **KoVAE**: https://github.com/azencot-group/KoVAE
- **KoonPro**: https://github.com/Rrh-Zheng/Koonpro
- **Koopa**: https://github.com/thuml/Koopa
- **MamKO**: https://openreview.net/forum?id=hNjCVVm0EQ
- **GeoTDM**: https://github.com/hanjq17/GeoTDM
- **EqMotion**: https://github.com/MediaBrain-SJTU/EqMotion
- **SympNets**: https://github.com/jpzxshi/sympnets
- **Symplectic-ODENet**: https://github.com/Physics-aware-AI/Symplectic-ODENet

## E. Limitation

We focus on stable and efficient long-horizon simulation within individual physical systems under benchmark-specific training and evaluation. A natural extension is to develop transferable geometric dynamics models that generalize across molecular compositions and broader dynamical regimes. Further empirical investigation of how step size and the geometric quantities appearing in our approximation bound affect model behavior remains an important direction.

## F. Concrete Two-Step Matrix Realization

We provide a concrete matrix realization of the discrete transport construction used in Section 4. For the natural action of a Lie group on $V$, the representation-dependent quantities for $\mathfrak{gl}(V)$ and $\mathfrak{o}(V, B)$ are summarized in Table 4.

*Table 4.* Matrix realizations of the transport ingredients under the natural action on $V$.

| Lie algebra | Constraint | Representation | $\rho_*(X)$ | $\mathrm{Ad}_g(X)$ | $\Phi(X \wedge Y)$ |
|---|---|---|---|---|---|
| $\mathfrak{gl}(V)$ | none | $\rho(g) = g$ | $\rho_*(X) = X$ | $gXg^{-1}$ | $\frac{1}{2}[X, Y]$ |
| $\mathfrak{o}(V, B)$ | $g^\top Bg = B, X^\top B + BX = 0$ | $\rho(g) = g$ | $\rho_*(X) = X$ | $gXg^{-1}, gXB^{-1}g^\top B$ | $\frac{1}{2}[X, Y]$ |

We now consider a two-step discretization under $\mathfrak{g} = \mathfrak{gl}(2)$, where $r = \dim(\mathfrak{gl}(2)) = 4$ and $\rho_*(X) = X$. Given a basis $\{E_a\}_{a=1}^4$, define two local Lie-algebra increments by

$$A_1 = h_1 \sum_{a=1}^4 \alpha_a(t_0, x_0; \theta)E_a, \qquad A_2 = h_2 \sum_{a=1}^4 \alpha_a(t_1, x_0; \theta)E_a,$$

and let

$$g_1 = \exp(A_1), \qquad g_2 = \exp(A_2).$$

The corresponding lifted increments are

$$\Delta S_1 = (g_1, (A_1, 0)), \qquad \Delta S_2 = (g_2, (A_2, 0)).$$

Applying the semidirect-product composition in Definition 4.1 yields the two-step cumulative state

$$\mathcal{S}_2 = \Delta S_1 \ltimes \Delta S_2 = \left( g_1 g_2, \ A_1 + \mathrm{Ad}_{g_1}(A_2), \ \frac{1}{2} A_1 \wedge \mathrm{Ad}_{g_1}(A_2) \right).$$

Therefore, the discrete transport operator associated with the update from $x_0$ to $x_2$ becomes

$$\Pi_2 := \widehat{\mathcal{PT}}^\rho_{0 \to t_2}$$

$$= \rho(g_1 g_2) \exp\left( \rho_*\left( \Phi\left( \frac{1}{2} A_1 \wedge \mathrm{Ad}_{g_1}(A_2) \right) \right) \right)$$

$$= \exp(A_1) \exp(A_2) \exp\left( \frac{1}{4} \left[ A_1, \exp(A_1) A_2 \exp(-A_1) \right] \right).$$

The propagated state is then given by

$$x_2 = \Pi_2 x_0.$$

This expression makes explicit how the first-order matrix increments are composed with the second-order commutator correction induced by the semidirect-product transport.

For $\mathfrak{o}(V, B)$, the same two-step construction applies with $A_1, A_2 \in \mathfrak{o}(V, B)$ and $g_1, g_2 \in O(V, B)$. In this case, $g_i^\top B g_i = B$, and the commutator correction remains in $\mathfrak{o}(V, B)$ because the orthogonal Lie algebra is closed under the bracket.

## G. Algorithmic Details

We describe the implementation of the graded Euler-Poincaré prefix scan that produces the cumulative generator sequence used in the latent evolution of Section D. At each timestep, the model outputs a Lie–algebra increment $\Delta u_m$, and these increments are accumulated according to the graded semidirect-product structure introduced in Section B.2. The procedure below implements this accumulation through an associative prefix scan, enabling parallel evaluation over long horizons. For each increment $\Delta u_m$ the algorithm initializes the graded components $(U_m, C_m, Z_m^{(3)}, \ldots, Z_m^{(R)})$ and the locally linearized adjoint operator $\mathrm{Ad}_m$. The COMBINE operator performs the graded semidirect-product composition, that transports all graded components through the adjoint action and aggregates them using the recursive wedge rules defined in Definition B.2, and the associativity of this operator permits its use within a standard parallel prefix-scan. Each cumulative element $A_m^{\mathrm{pref}}$ is then mapped to a generator $\Theta_m$ via the curvature map $\Phi$, which collects contributions from all antisymmetric orders, resulting sequence $\{\Theta_m\}$ parametrizes the truncated matrix exponential used in the latent update. The overall computational cost is $\mathcal{O}(M)$ work with $\mathcal{O}(\log M)$ sequential depth. The pseudocode for this algorithm is given in Algorithm 1.

---

**Algorithm 1** Parallel Euler–Poincaré prefix scan

---

**Require:** Per-step Lie algebra increments $\{\Delta u_m\}_{m=0}^{M-1}$, structure constants $C$, truncation order $R$
**Ensure:** Cumulative generators $\{\Theta_m\}_{m=0}^{M-1}$
  {Initialize graded elements $(U, C, Z^{(3)}, \ldots, Z^{(R)})$}
  **for** $m = 0$ **to** $M - 1$ **do**
    $U_m \leftarrow \Delta u_m$
    $C_m \leftarrow 0$
    **for** $r = 3$ **to** $R$ **do** $Z_m^{(r)} \leftarrow 0$ **end for**
    $\mathrm{ad}_m \leftarrow \sum_k (U_m)_k \, C_k$
    $\mathrm{Ad}_m \leftarrow I + \mathrm{ad}_m + \frac{1}{2}\mathrm{ad}_m^2$
    $A_m \leftarrow (U_m, C_m, Z_m^{(3)}, \ldots, Z_m^{(R)}, \mathrm{Ad}_m)$
  **end for**
  {Graded semidirect-product composition}
  **function** COMBINE($A^{(1)}, A^{(2)}$)
    $(U_1, C_1, Z_1^{(3)}, \ldots, Z_1^{(R)}, \mathrm{Ad}_1) \leftarrow A^{(1)}$
    $(U_2, C_2, Z_2^{(3)}, \ldots, Z_2^{(R)}, \mathrm{Ad}_2) \leftarrow A^{(2)}$
    {Transport all graded components}
    $\tilde{U}_2 \leftarrow \mathrm{Ad}_1 U_2$
    $\tilde{C}_2 \leftarrow (\wedge^2 \mathrm{Ad}_1) C_2$
    **for** $r = 3$ **to** $R$ **do**
        $\tilde{Z}_2^{(r)} \leftarrow (\wedge^r \mathrm{Ad}_1) Z_2^{(r)}$
    **end for**
    {Accumulate each graded level}
    $U \leftarrow U_1 + \tilde{U}_2$
    $C \leftarrow C_1 + \tilde{C}_2 + \frac{1}{2}(U_1 \wedge \tilde{U}_2)$
    **for** $r = 3$ **to** $R$ **do**
        $Z^{(r)} \leftarrow Z_1^{(r)} + \tilde{Z}_2^{(r)}$
        **for** $a = 1$ **to** $r - 1$ **do**
            **if** $a = 1$ **then** $Z^{(r)} \leftarrow Z^{(r)} + \frac{1}{r}(U_1 \wedge \tilde{Z}_2^{(r-1)})$
            **else if** $a = 2$ **then** $Z^{(r)} \leftarrow Z^{(r)} + \frac{1}{r}(C_1 \wedge \tilde{Z}_2^{(r-2)})$
            **else** $Z^{(r)} \leftarrow Z^{(r)} + \frac{1}{r}(Z_1^{(a)} \wedge \tilde{Z}_2^{(r-a)})$
        **end for**
    **end for**
    $\mathrm{Ad} \leftarrow \mathrm{Ad}_1 \mathrm{Ad}_2$
    **return** $(U, C, Z^{(3)}, \ldots, Z^{(R)}, \mathrm{Ad})$
  **end function**
  {Parallel prefix scan}
  $\{A_m^{\mathrm{pref}}\} \leftarrow$ PARALLELSCAN($\{A_m\}$, COMBINE)
  **for** $m = 0$ **to** $M - 1$ **do**
    $(U_m, C_m, Z_m^{(3)}, \ldots, Z_m^{(R)}, \_) \leftarrow A_m^{\mathrm{pref}}$
    $\Theta_m \leftarrow U_m + \Phi(C_m, Z_m^{(3)}, \ldots, Z_m^{(R)})$
  **end for**
  **return** $\{\Theta_m\}$

---

