# OpenReview forum: "Euler–Poincaré Neural Dynamics: A Geometric-Mechanics Framework for Scientific Simulation"
_ICML.cc/2026/Conference — ICML 2026 regular_

### Official Review · Reviewer_q9PR · 2026-03-04

**Soundness:** 3
**Presentation:** 2
**Significance:** 3
**Originality:** 4
**Overall Recommendation:** 5
**Confidence:** 3

**Summary:**

The authors propose a data-driven simulation approach for unrolling physical systems. The key idea is to compute the evolution of a system in the latent space where the motion of an object is dictated by Lie algebra. A low-dimensional representation of an initial state is computed first and it plays a role of a conditioning vector for the neural field which predicts coefficients of the action (defined in a Lie algebra specific basis) at every moment of time. Once the coefficients are generated, the trajectory is integrated efficiently via parallel scan. Because the motion is constrained to the basis of Lie algebra, the resulting states never violates the geometric constraint thus remaining geometrically valid during simulation. The method is validated on protein simulation and electrolyte hydration where it demonstrates considerably stronger performance compared to baselines.

**Compliance With Llm Reviewing Policy:**

Affirmed.

**Final Justification:**

My primary concern with the paper was the scope of their theoretical claims which were, as authors agreed, overstated. As this aspect was sorted out, I believe the submission has sufficient originality and quality to be presented at the conference, with a potential for further work. I am not willing to give it Strong Accept for the limited impact, but I consider Accept to be a fair score.

**Key Questions For Authors:**

- How much of a problem is that the method is not rotationally equivariant?
- Did you measure the equivariance error? Does it anyhow follow from the method (since it is claimed to be geometrically consistent), or does one need to use special encoder (e.g. as in equivariant neural field) to enforce it?
- Have you though what it would take to achieve transferability of the method? I.e. if I have multiple proteins, is there a way to adapt the method to simulate them or is the method inherently yields system-specific solutions?

[1] Grounding Continuous Representations in Geometry: Equivariant Neural Fields, David R Wessels, David M Knigge, Samuele Papa, Riccardo Valperga, Sharvaree Vadgama, Efstratios Gavves, Erik J Bekkers

**Limitations:**

yes

**Strengths And Weaknesses:**

__Soundness__
- The submission is technically sound.
- The paper mentions guaranteed physical consistency multiple times, although I am not convinced that it holds. First of all, what is physical consistence in this context? I would assume it must refer to conservation laws or conservation of the true Hamiltonian the real-world system, which, as far as I understand, doesn't exactly follow from satisfying the Euler-Arnold equation *in the latent space*. I would appreciate if authors could clarify what they mean here specifically.
- The paper mentions that the solution strictly follow the Euler–Arnold equation, but is that the case? Is it guaranteed that using NN outputs results in a perfect solution of the equation?
- I am not sure about the claim of mitigating stepwise error accumulation over long horizons. As far as I get it, it is linear and not exponential, which is still remarkable, but it is not exactly what I would count as error mitigation.
- I believe that the lack of transferability is a major drawback of the approach as it is now, as a model is only valid for a single system for which it is trained. While it does not reduce the contribution, I would argue that it has to be mentioned.
- The method is neither rotational nor permutation equivariant, which is another drawback that is not discussed. The encoder is an MLP which essentially flattens a state. Permutation equivariance will be an issue should authors pursue transferability of their method, and rotation equivariance seems like an important property for the systems considered in the paper. Did authors test equivariance error?
- Authors claim interpretability, but dimensions of the latent Lie algebra are opaque, how exactly does interpretability work here?

__Presentation__
- The paper is extremely tense mathematically. I am glad authors provide figures and visual highlights to remedy that.
- Overall, I am satisfied with the presentation, but I would love to see more simple language interpretation, perhaps even at the cost moving parts of math to the appendix.

__Significance__
- I believe that the method suggested in the paper might lay a foundation for very long simulations, which would be extremely valuable for computational chemistry and physics.
- Long horizon simulation of specific systems in the latent space is already a remarkable achievement.
- Reversibility of the framework might be practical for inverse problems.

__Originality__
- The paper is an original piece of work that introduces a novel method of simulating a physical system in a geometrically consistent way.
- Authors propose an elegant trick of parallel scan to unroll a trajectory on sub-linear time, which is an impressive property.
- Contributions are sound and are clearly distinguished from the existing literature.

---

> ### Author Rebuttal · Authors · 2026-03-30
>
> We sincerely thank you for the positive assessment of our work and for the thoughtful and constructive questions. We greatly appreciate your careful engagement with the manuscript, and we found the feedback very helpful in identifying where the paper can be clarified and strengthened. We take these concerns seriously and will address them more explicitly in the revision. We hope that these clarifications and revisions will adequately resolve the concerns raised and support an even more favorable view of the paper.
>
> **Q. what is physical consistence in this context...**
>
> We thank you for this important question. In our paper, physical consistency refers more specifically to whether the generated trajectories stably preserve physically meaningful structural observables in the real molecular coordinate space over long horizons. In our experiments, this is supported by metrics such as $C_\alpha$-RMSD. For example, Figure 5 shows that the RMSD remains stable over long-time horizons for both IVP and TVP settings.
>
> **Q. The paper mentions that the solution strictly follow…**
>
> Thank you for the thoughtful question. We do not claim that arbitrary NN outputs automatically satisfy the Euler–Arnold equation. Rather, in the continuous formulation, Proposition 3.1 shows that if the coefficients $\alpha_a(t,z;\theta)$ satisfy Eq. (12), then the induced field $\xi_\theta(t) = \sum_a \alpha_a (t,z;\theta)E_a$ follows the Euler-Arnold or geodesic equation by construction. This is derived in Appendix B.1 using Proposition B.2 and the proof of Proposition 3.1. In practice, our scan-based rollout approximates this equation with NN outputs.
>
> **Q. I am not sure about the claim of mitigating stepwise error…**
>
> The relevant point here is not exact linearity of rollout error. The method does not recursively feed predicted states back into the model, but instead predicts Lie-algebra increments over the horizon and composes them through a parallel prefix-scan. It avoids the autoregressive feedback compounding that often leads to unstable long-horizon rollout in sequential methods. In this sense, our claim is that the proposed formulation mitigates this specific source of stepwise error amplification.
>
> **Q. I believe that the lack of transferability…Have you though what it would take…**
>
> Thank you for this important observation. Our current method does not address cross-system transfer and should be viewed as a system-specific framework. The goal of our work is stable and efficient long-horizon simulation within a given physical system, and the experiments are designed accordingly. We recognize that transferability is an important direction for broader applicability, and we will discuss this point in the revision.
>
> **Q. Authors claim interpretability, but dimensions of…**
>
> Thank you for this thoughtful question. In our paper, interpretability was intended in a structural sense. We parameterize the dynamics as Lie-algebra velocities through neural coefficients, convert them into discrete group increments, and compose them through a discrete parallel transport rollout, rather than modeling evolution as an unconstrained black-box map. In particular, Proposition 4.2 provides an error analysis for this construction, which is why we viewed the model as more structurally analyzable and transparent. We understand that the term interpretability may have suggested a stronger claim than we intended. We apologize for that confusion, and will revise the wording accordingly.
>
> **Q. The method is neither rotational nor permutation…How much of a problem…Did you measure the equivariance error?…**
>
> We would like to clarify that the role of Lie structure in our work differs from that in prior equivariance based methods. In those approaches, Lie groups or related group structure are typically used to encode input space symmetries such as rotations or permutations. In contrast, in our setting Lie groups are introduced as a natural framework for lifting the time evolution of geometric dynamics into an associative form. As a result, the primary focus of our method is not equivariance, but associative composition and parallel scan for efficient long horizon rollout computation. This temporal acceleration mechanism yields $\underline{300\times}$ speedups compared to equivariance based methods, which is a goal distinct from that of existing equivariance based approaches. Moreover, our model is not designed to satisfy equivariance, but this is not the factor that governs performance.

---

> > ### Author Rebuttal · Reviewer_q9PR · 2026-04-02
> >
> > I appreciate authors time and engagement. I am willing to raise my score to Accept (5) under the condition that the wording in the points we discussed will be revised (physical consistency, error mitigation, interpretability).

---

> > > ### Author Response · Authors · 2026-04-05
> > >
> > > We thank the reviewer for the thoughtful follow-up and the positive reassessment of our work. We agree that the wording regarding physical consistency, error mitigation can be made more precise. In the final manuscript, we will revise the relevant wording to clarify the intended scope of each claim and to rephrase any potentially misleading expressions with greater care. We are grateful for this constructive and helpful feedback.

---

### Official Review · Reviewer_FftU · 2026-03-11

**Soundness:** 3
**Presentation:** 2
**Significance:** 2
**Originality:** 3
**Overall Recommendation:** 3
**Confidence:** 3

**Summary:**

The paper proposes an operator learning framework rooted in geometric mechanics. When the dynamics to be learned take place in a Lie algebra, the proposed method leads to a model that automatically adheres to the system's symmetries and physical invariants. Furthermore, a parallel computation strategy is proposed to reduce the computational cost in both training and rollout. The paper offers an overview over the theoretical framework in a general setting and shows numerical examples from protein folding and electrolyte hydration systems.

**Compliance With Llm Reviewing Policy:**

Affirmed.

**Final Justification:**

I am still hesitant to suggest this paper for acceptance simply because I cannot judge how the camera-ready version will read after the significant changes that were discussed.

I would like to emphasize that my qualms are not mainly with the notation and theory itself, but that I feel that not a lot of effort has been made to "translate" between the paper's content and more widely-known concepts. Without this translation effort, one can simply omit a lot of definitions from the paper and suggest the reader to read the paper together with a geometric mechanics textbook (Abraham/Marsden or Holm) on the side. In the rebuttal chain to my review, the authors provided a translated version of the proposed method for one of the examples from the paper, which I want to acknowledge. The authors made a sincere effort to address my concerns. I have raised my score accordingly.

**Key Questions For Authors:**

1) What is $\mathfrak{g}$ in the experiments from Section 6?
2) Is it possible to describe the proposed method for the examples from Section 6 purely in the form of matrix ODEs?
3) In Proposition 4.2, the last term does not converge to zero as $h \to 0$. Is this not a problem?

**Limitations:**

The numerical experiments seem to be rather small-scale; I am not an expert in molecular dynamics simulations, but I was under the impression that these are often run on very large scale hardware. Can you comment on this?

**Strengths And Weaknesses:**

Soundness: The paper thoroughly introduces the concepts from geometric mechanics it draws from and provides the computations to derive them in the appendix. The numerical experiments are thoroughly described in the appendix and the authors put effort in ensuring that baselines of competing methods are evaluated under the same conditions; this I want to commend. There only few typos and inaccuracies I could find (e.g. the inertia operator is introduced twice, once mapping to the algebra, once to its dual). I consider this a sound paper, albeit much of the mathematics is hidden in the heavy notation typical for the field of geometric mechanics.

---

Presentation: The heavy notation is the main challenge in understanding the paper. I am not convinced the generality of the setting is helpful for the exposition. For example, I am until now not sure what Lie algebra the dynamics from Section 6 take place in (is it the generalized Orthogonal Group?). The Koopman framework is introduced with general observables $f$, but from what I understand the only observable considered is the identity $\iota$. My suggestion is to pick a concrete example early on (e.g. the one from Fig. 1) and guide the reader through the text with it. The parallel scan (Appendix C) is a core contribution of the paper and might warrant more space in the main text.
Overall, I frankly find the paper too focussed on introducing a lot of geometrical concepts of notation that an ICML audience is unlikely to already be familiar with and I am not convinced that the amount of jargon and notation is helping to explain the proposed method.

---

Significance and Originality: The problem considered - operator inference that is aware of geometrical constraints - is certainly significant. The extent to which the paper advances understanding and the state of the art requires to separate new methodology from new notation. I find this hard to judge. It would be easier for me to do so with a more concrete explanation of the algorithm in the main text, i.e. moving it up from page 28 and embedding it in the exposition. The numerical results show an improvement over the competing baseline methods on nearly all metrics. Given that the experiments seem meticulously done, this is strong evidence for impactful work.

---

> ### Author Rebuttal · Authors · 2026-03-30
>
> We thank you for the careful reading and the thoughtful, detailed feedback. During the preparation of this manuscript, we were aware that the mathematical exposition might feel challenging. In the manuscript, we tried to prioritize the mathematical formality and structural rigor of the proposed method. We fully acknowledge the reviewer’s concern that this may have limited the intuitive accessibility of the core ideas.
>
> We were very encouraged by your positive remarks on the technical strengths of our work. We fully recognize that the initial presentation fell short in making the core ideas as accessible as they should have been, and we have carefully followed your suggestions to improve the exposition. We hope that the clarifications below and the concrete revisions we outline will help address your main concerns and support a more favorable overall assessment.
>
> **Q. The Koopman framework is introduced with...**
>
> We recast state evolution as geometric transport through the identity observable $\iota$ in our approach, which is a specialized case of the general observables $f$, and not intended to address $f$ itself. We will clarify early in the Introduction and Method sections that the general Koopman setting serves primarily as motivation.
>
> **Q. My suggestion is to pick…**
>
> We will introduce, earlier in the Method section, the same concrete $\mathfrak{gl}(N)$-based running example shown in Fig.1 and use it as a guiding example throughout the presentation, so that the abstract construction is developed in direct one-to-one correspondence with the figure. More specifically, we will revise the exposition to integrate the text more tightly with Fig.1, presenting the example from a unified perspective that connects the Koopman dynamics, the associated $\mathfrak{gl}(N)$-valued matrix ODE, and the induced Lie-group evolution.
>
> **Q. What is $\mathfrak{g}$…**
>
> In practice, we used $\mathfrak{g}$ in Table 1 for experiments, namely the general linear Lie algebra and orthogonal Lie subalgebra.
>
> **Q. Is it possible to describe .. form of matrix ODEs ..?**
>
> As a concrete example, we describe the two-step discretization of the matrix ODE $x_0 \rightarrow x_2$ as
> $\Pi_2=\rho\\!\left(\exp\\!\left(h_1\sum_{a=1}^{r}\alpha_a(t_0,z_0;\theta)E_a\right)\exp\\!\left(h_2\sum_{b=1}^{r}\alpha_b(t_1,z_0;\theta)E_b\right)\right)\cdot\exp\\left(\rho_*\\!\left(\Phi\ \left(\frac{h_1h_2}{2}\left(\sum_{a=1}^{r}\alpha_a(t_0,z_0;\theta)E_a\right)\wedge\mathrm{Ad}\_{\exp\ \left(h_1\sum_{c=1}^{r}\alpha_c(t_0,z_0;\theta)E_c\right)}\ \left(\sum_{b=1}^{r}\alpha_b(t_1,z_0;\theta)E_b\right)\right)\right)\right) $
>
> $z_2 = \Pi_2 z_0, \qquad x_2 = \mathrm{Dec}_{\phi}(z_2), \qquad z_0 = x_0.$
>
> This provides an equivalent matrix-ODE interpretation of $\dot{\mathbf{x}}\_t=\rho_\*\(\dot{\xi}_{\theta}(t))\mathbf{x}_t$,
> which arises naturally from the semidirect-product structure in Def 4.1 together with the parameterization of the geodesic coefficients in Eq.(12). For brevity, we do not include the full derivation in the rebuttal, but we will clarify this connection more explicitly in the revision. We also note that, although it admits a matrix-ODE description, the actual computation is not performed by directly integrating ODEs. Instead, the model uses the scan-compatible closed-form propagation shown in the blue box on page 5 and in Eq.(16).
>
> **Q. The parallel scan (Appendix C) is… It would be easier for me to do so…**
>
> In the current draft, the geometric language and notation may obscure the parallel scan construction and its contribution. We will move the core content of the parallel scan, now in Appendices C and E, into Section 4. Concretely, we will add a compact algorithm box that separates the scan into three parts, namely the inputs and outputs, the upsweep/downsweep, and the parallel computations. We will also state that this algorithm reduces the rollout time complexity from $O(M)$ to $O(\log M)$. This revision will keep the geometric notation for interpretation while making the scan-based acceleration more visible in the main text.
>
> **Q. In Proposition 4.2, the last term…**
>
> The remaining $diam~G$ term is a predetermined constant that stays fixed and is much smaller than other quantities, such as $|\dot\xi|$, during training. Therefore, it is negligible in practice and does not pose a practical concern.
>
> **Q. The numerical experiments seem to be…**
>
> We agree that molecular dynamics simulations are often carried out at very large scale on dedicated HPC systems. Our experiments are not meant to replicate the largest such campaigns. Instead, we focus on a setting where the reference simulations are already expensive enough to make neural modeling meaningful. The dataset required roughly two weeks to generate on a single RTX A6000. In addition, the scale of our experiments is already about 20-100× larger than those considered in prior works. We will clarify this point in the revision so that the computational setting is better contextualized.

---

> > ### Author Rebuttal · Reviewer_FftU · 2026-04-02
> >
> > Thank you for your thorough response. I am selecting option (c) here reluctantly because I asked for very substantial modifications to the presentation of the paper. I trust that this is possible and appreciate your openness to do so.
> >
> > Maybe the Koopman framework exposition can be moved to get the space needed to do so. I am still not sure if the Koopman viewpoint is necessary to present the method which ultimately uses $f = \iota$.
> > ___
> > *Proposition 4.2*: Thank you for the clarification.
> > ___
> > *Scale*: Is the number 20-100x referencing other structure-preserving approaches or another subset of existing work that excludes the HPC campaigns?
> > ___
> > *Presentation*: If I wanted to implement this method for the case of the general linear Lie algebra $\mathfrak{gl}$, I believe to understand that:
> > - $\rho_*$ is the identity.
> > - $\mathrm{Ad}_{\exp A} B$ is a product between matrix exponentials of $A$ and $B$.
> > - $\Phi( \dots \wedge \dots)$ is a matrix commutator.
> > Is this accurate? In light of your response, it might be more natural to show the worked example ($\mathfrak{gl}$ or $\mathfrak{o}$) for the blue box on page 5. This (very concrete) example would also make it much easier to understand the differences to existing operator learning approaches that are not described in the geometric language employed by the paper.
> > ___
> > Once again, thank you for your patience and explanations. I understand my raised points are hard to respond to in the rebuttal process and I will continue to monitor the discussion here and with the other reviewers.

---

> > > ### Author Response · Authors · 2026-04-05
> > >
> > > Thank you again for the thoughtful and detailed follow-up. We are very grateful for the care with which you read the paper and for the constructive spirit of your comments. In particular, your feedback on the paper’s organization, flow, and overall readability was extremely valuable, as it helped us identify concrete ways to present the ideas and contributions more accessibly for readers. We hope that the clarifications below will better communicate the central ideas and contributions, and we would be sincerely grateful if they could be taken into consideration in your final evaluation.
> > >
> > > We introduced the Koopman perspective primarily as a familiar entry point for readers from the ML operator-learning community. We recognize, however, that this discussion may receive more emphasis than its role in the paper requires, which can affect readability. Since the central construction is developed through the Lie-algebraic formulation and does not rely as heavily on the broader Koopman formalism, we believe the presentation could be improved by making the Koopman discussion more concise and moving part of it to the related work and appendix. This would allow readers to reach the main idea more directly.
> > >
> > > The 20–100x statement is meant relative to prior ML studies on small-scale molecular dynamics, including both structure-preserving and generative trajectory approaches, rather than to large HPC campaigns or production-scale MD simulations.
> > > On the structure-preserving side, PAPS [1] is a recent example that reports molecular dynamics experiments on small molecules such as methane and aspirin. Among generative trajectory models, GeoTDM is evaluated in the small-molecule regime of MD17, which consists of eight molecules ranging from 9 to 21 atoms. In contrast, our experiments consider all-atom peptide systems, namely 20-residue TRP-CAGE and 35-residue Villin, containing 284 and 582 atoms, respectively. Our point is that these systems are substantially larger than the settings commonly considered in ML studies and also clearly beyond the MD17 small-molecule regime. We will revise the manuscript to make this comparison framework explicit.
> > >
> > > | Group | Representative work | Evaluation system | Experimental scale |
> > > | --- | --- | --- | --- |
> > > | Structure-preserving | PAPS | methane and aspirin | 5 and 21 atoms |
> > > | Generative trajectory | GeoTDM | MD17 8 molecules | 9 to 21 atoms |
> > > | Ours | Our method | TRP-CAGE, Villin | 284 and 582 atoms |
> > >
> > >
> > >
> > > For $\mathfrak{gl}(V)$ and $\mathfrak{o}(V,B)$, we summarize in the table below the corresponding expressions for $\rho$, $\rho_*$, $\mathrm{Ad}_g$, and $\Phi(X \wedge Y)$. Together with the earlier clarification, we hope this makes it easier to see how the Lie algebra notation in the current presentation translates into a concrete matrix ODE form. Following the reviewer’s suggestion, we will also use the concrete example provided below the table as a short worked example near the blue box, together with a brief specialization to the orthogonal case.
> > >
> > > | Lie algebra | Constraint | Representation on $V$ | $\rho_*(X)$ | $\mathrm{Ad}_g(X)$ | $\Phi(X \wedge Y)$ |
> > > |---|---|---|---|---|---|
> > > | $\mathfrak{g}=\mathfrak{gl}(V) $ | none | $\rho(g) = g$ | $\rho_*(X) = X$ | $\mathrm{Ad}_g(X)=gXg^{-1}$ | $\Phi(X\wedge Y)=\tfrac12[X,Y]$ |
> > > | $\mathfrak{g}=\mathfrak{o}(V,B) \quad$ | $g^\top B g = B, \\ X^\top B + BX = 0 \quad $ | $\rho(g) = g$ | $\rho_*(X) = BX \quad $ | $\mathrm{Ad}_g(X)=gXg^{-1}=gXB^{-1}g^\top B \quad $ | $\Phi(X\wedge Y)=\tfrac12[X,Y]$ |
> > >
> > > As a concrete example, we consider the two-step discretization from $x_0$ to $x_2$ under the $\mathfrak g=\mathfrak{gl}(2)$. In this case, the structured propagation is represented by the discrete transport operator $\Pi_2$, while the conventional formulation is represented by the learned evolution operator $\mathcal U_{s,t}^{\theta}$.
> > >
> > > Instantiating Table 1 with the general linear Lie algebra $\mathfrak g=\mathfrak{gl}(2)$, for which $r=4$ and $\rho_{*}(X)=X$, we can write the two-step discretization of the matrix ODE $x_0 \to x_2$ as
> > >
> > > $$
> > > \Pi_2=\exp\\!\big(h_1\sum_{a=1}^{4}\alpha_a(t_0,z_0;\theta)E_a\big)\exp\\!\big(h_2\sum_{b=1}^{4}\alpha_b(t_1,z_0;\theta)E_b\big)\cdot\exp\\!\Bigg(\frac{h_1h_2}{4}\Bigg[\sum_{a=1}^{4}\alpha_a(t_0,z_0;\theta)E_a,\ \exp\\!\big(h_1\sum_{c=1}^{4}\alpha_c(t_0,z_0;\theta)E_c\big)\Big(\sum_{b=1}^{4}\alpha_b(t_1,z_0;\theta)E_b\Big)\exp\\!\big(-h_1\sum_{c=1}^{4}\alpha_c(t_0,z_0;\theta)E_c\big)\Bigg]\Bigg),
> > > $$
> > >
> > > $$
> > > z_2=\Pi_2 z_0,\~x_2=\operatorname{Dec}_{\phi}(z_2),~z_0=x_0.
> > > $$
> > >
> > > ---
> > >
> > > [1] : S Park and J Kim. Poisson-Algebraic Parallel Scan: A Fast Symplectic Framework for Neural Hamiltonians. OpenReview, 2025, https://openreview.net/pdf?id=ZjZo4h80XL.

---

### Official Review · Reviewer_bKkE · 2026-03-12

**Soundness:** 3
**Presentation:** 3
**Significance:** 2
**Originality:** 4
**Overall Recommendation:** 5
**Confidence:** 3

**Summary:**

I first off want to say that this submission is quite similar to 3736 and 4033 in both figures, tables and references and the initial setup.
The core ideas explored in both works are different, but nonetheless I believe that the authors should cross reference their work as per the dual submission policy and highlight the differences.

This work attempts to perform operator learning via geodesics on a Lie group derived from the Euler–Arnold equations under a left-invariant metric, and then transports states using a parallel transport operator induced by a group representation.

A distinctive aspect of this work is that the authors define both forward transport and a reversed-time transport from the same geometric objects. This enables one to solve a terminal value problem and reconstruct earlier states from terminal observations without retraining.

For scalability, the authors introduce an extended “parallel transport group” constructed as a semidirect product over an extended algebra that includes wedge/exterior terms, so that discrete updates accumulate both group motion and curvature-related corrections in an associative way. This enables an prefix scan to reduce computation costs.

The experiments emphasize long-horizon molecular system dynamics, and the aforementioned new capability of stable reverse evolution.

**Compliance With Llm Reviewing Policy:**

Affirmed.

**Final Justification:**

The authors have addressed my comments, and I have raised my score as a result.

**Key Questions For Authors:**

(apologies for the lack of latex formatting)

Line 102, right column. Define is $g_\epsilon$.

Line 95 right column and line 133 left colum: $\mathbb{I}$ is defined both as $\mathbb{I}:\mathfrak{g}\to\mathfrak{g}$ and $\mathbb{I}:\mathfrak{g}\to\mathfrak{g}^*$

I'm also confused about the notation of ad_{\xi}.
In (6), ad_{\xi}^* has a star, and is defined as \langle {ad}_{\xi} m,\eta \rangle_\mathfrak{g}= \langle m,{ad}_{\xi}\eta\rangle_\mathfrak{g}. First off, it says "for all $\eta\in\mathfrak{g}$" but this must surely also be for all $m$? Then in 150/151, $X, Y$ are not defined, and more confusingly, the definition of $\mathrm{ad}^{\dagger}$ coincides with that $\mathrm{ad}_{\xi}^*$, although now with respect to unintroduced $u,v$. Maybe I'm missing something but this seems like different styles of notation are being mixed.

Equation (8). The parallel transport is obtained from Eq(7) since it is the solution of the $\dot\xi(t) = -\mathrm{ad}^{\dagger}_\xi (\xi)$? Please add some more clarification here.

Equation (9), why is there a minus sign in \xi^\#(t) =-\xi(T-t)? Should it not be \xi^\#(t) =\xi(T-t) based on the definition of \# above?

With regards to the numerical experiments, what $\mathfrak{g}$ is chosen here? Table 1. lists the structure constants of various Lie algebra's, but it's not clear which ones are  used in the experiments.

Can the authors motivate the modeling assumptions of their choice of $\mathfrak{g}$ for the reported experiments? Why would one Lie algebra be better than others?

What are the units of the inference time and training time reported for the experiments?

Finally, where can the code be found to reproduce these results? I see that the baseline code is mentioned in the appendix, but where can the author's code be found?

**Limitations:**

The authors do not discuss the limitations of their work in the conclusion.

**Strengths And Weaknesses:**

The results seem correct and the claims are well supported. The results are understandable despite the somewhat technical exposition.
It is hard to judge the significance of the experiments for me, since I'm not fully immersed in the state-of-the-art of this area of research.
It does seem that the method is novel, and really appreciate the differentiable geometric approach to this subject, which is quite beautiful.

---

> ### Author Rebuttal · Authors · 2026-03-30
>
> Thank you for your careful reading and thoughtful feedback. We sincerely appreciate your positive assessment of the paper’s technical soundness and novelty. We also agree that several points would benefit from clearer presentation. To address your concerns and further improve the overall quality of the manuscript, we will revise the relevant passages accordingly. We hope that these revisions will make the paper clearer and that our response has been helpful in addressing your comments.
>
> **Q. notation issues**
>
> Regarding the notation issues, we would like to respond as follows.
>
> 1. Line 102, right column: $g_\epsilon$
>
> In the current manuscript, we introduced $\delta g(t)=\left.\frac{d}{d\epsilon}\right|\_{\epsilon=0}g\_\epsilon(t)$ without first defining $g_\epsilon(t)$. In the revised manuscript, we will define $g_\epsilon(t)$ at its first occurrence as a smooth one-parameter family satisfying $g_0(t)=g(t)$.
>
> 2. Line 95, right column and line 133, left column
>
> You are correct. Our intended notation is $\mathbb{I}:\mathfrak{g}\to\mathfrak{g}^\*$ and $m:=\mathbb{I}(\xi)\in\mathfrak{g}^\*$. Any place in the current manuscript where $\mathbb{I}:\mathfrak{g}\to\mathfrak{g}$ or $m\in\mathfrak{g}$ could be inferred is due to inconsistent notation. In the revised version, we will consistently use $\mathbb{I}:\mathfrak{g}\to\mathfrak{g}^\*$ and $m\in\mathfrak{g}^\*$.
>
> 3. $ad_\xi^*$ and $ad_\xi^\dagger$
>
> Our intention is that $\mathrm{ad}^\*\_\xi$ denotes the coadjoint action on $\mathfrak g^\*$, whereas $\mathrm{ad}\_\xi^\dagger$ denotes the metric adjoint on $\mathfrak g$. We will revise the text to make this distinction explicit, including the correct quantification $\forall m\in\mathfrak g^\*, \forall \eta\in\mathfrak g$ in the definition of $\mathrm{ad}_\xi^\*$. We will also state that $\operatorname{ad}\_\xi^\dagger=\mathbb I^{-1}\operatorname{ad}^\*\_\xi\mathbb I$, and introduce $X,Y,u,v\in\mathfrak g$  before using them, so that the notation is consistent throughout.
>
> 4. Equation (8)
>
> Equation (8) is the solution to the ODE $\frac{d}{dt}PT^\rho_{s\to t}=-\rho_*(ad^\dagger_{\xi(t)})PT^\rho_{s\to t}$ with initial condition $PT^\rho_{s\to s}=Id$.
>
> 5. Equation (9)
>
> This expression is correct as written. If $g^\sharp(t)=g(T-t)$, then by the chain rule, $\xi^\sharp(t)=(g^\sharp(t))^{-1}\dot{g}^\sharp(t)=-\xi(T-t)$.
>
> **Q. With regards to the numerical experiments, what $\mathfrak{g}$ is chosen here?… Why would one Lie algebra be better than others?**
>
> Thank you for these questions. In the experiments, we instantiated the framework using $\mathfrak g$ in Table 1, specifically the general linear Lie algebra and the orthogonal Lie subalgebra. Since the manuscript presents the method at the level of a general Lie algebra, these concrete experimental choices were not made as explicit as they should have been.
>
> Our goal in this work is not to claim that one Lie algebra is universally superior to others, nor to provide a comprehensive comparison across different Lie algebras. Rather, we first formulate the framework at the level of a general Lie algebra and then instantiate it with canonical and widely used examples. In this sense, these Lie algebras serve as natural representative choices for validating the general framework. We expect different Lie algebras to involve different trade-offs in expressivity, computational efficiency, and structure of the search space.
>
> **Q. What are the units of the inference time and training time...**
>
> The training time and inference time reported in the experimental tables are both measured in seconds. We will make this clear in the main text or in the table captions.
>
> **Q. Finally, where can the code be found to reproduce...**
>
> Regarding code availability, we did not include our code repository in the current double-blind submission. We plan to release the code and experimental results publicly upon acceptance.

---

> > ### Author Rebuttal · Reviewer_bKkE · 2026-04-02
> >
> > My questions have been adequately resolved, except for the lack of discussion about the limitation of their work and the connection with the other submissions I've seen.
> >
> > Given the strong similarities to 3736 and 4033, can the authors include a discussion about the different trade-offs that these three works represent?
> >
> > With regards to the code: if the paper is accepted I expect the code to be online before the camera-ready deadline.

---

> > > ### Author Response · Authors · 2026-04-05
> > >
> > > For clarity, in the discussion below, we refer to paper 3735 as EPND, paper 3736 as the Lie-Algebra paper, and paper 4033 as SPS.
> > >
> > > We thank the reviewer for noting the apparent similarity among the three manuscripts. We agree that they are connected in the sense that they belong to a common research agenda. Taken together, they form part of a broader program on mathematically structured operator learning, whose aim is to replace unconstrained black-box temporal propagation with composition laws that remain stable and computationally efficient over long horizons.
> > >
> > > That said, the overlap is only at this meta-level. The three papers are **fundamentally different** in their mathematical backbone, admissible dynamical regime, and scientific objectives. In particular, they are organized around three distinct mathematical cores, namely **Riemannian geometry** in EPND, **symplectic geometry** in SPS, and **Lie algebra** in the Lie-algebraic Koopman framework.
> > >
> > > For EPND, geometry itself is the main computational object. The method is built around Riemannian and geometric-mechanics tools such as geodesic flows, parallel transport, curvature-aware composition, and reversible forward/backward scans. This is precisely what makes the framework qualitatively different. It exposes an intrinsic reversible structure in the learned dynamics, which in turn enables terminal value reconstruction and other inverse problems that are difficult to formulate in conventional operator-learning pipelines. In that sense, EPND is not merely an acceleration method but is a geometry driven formulation of scanable dynamics. At the same time, this also makes it structurally constrained, since the method relies on geometry-compatible assumptions and is therefore not the most general engineering template.
> > >
> > > By contrast, SPS is the most constrained of the three works. It specializes from general operator learning to Hamiltonian systems, and in particular to a Poisson-algebraic construction of symplectic flows. Its main purpose is not general time-series forecasting, but the direct modeling of physically meaningful dynamics such as quantum dynamics/molecular dynamics with exact structural preservation, especially symplecticity, energy consistency. For this reason, SPS is naturally suited to scientific simulation where a genuine Hamiltonian structure exists, but it is not intended as a drop-in framework for generic forecasting benchmarks such as the ETT benchmarks.
> > >
> > > Finally, the Lie-algebraic Koopman work is the most engineering-oriented and the broadest in scope. The main goal is to make accelerated operator learning practical for broad classes of dynamical data by restricting generators to computationally favorable Lie subalgebras and composing finite-time propagators through an associative prefix scan. The central emphasis is therefore on the accuracy-efficiency tradeoff, asking which Lie subalgebra yields the most favorable computational profile and structural bias for a given problem class while still supporting scalable parallel composition. In this sense, its contribution is not exact physical invariance as in SPS, nor reversible geometric inversion as in EPND, but an algebraically structured and computationally optimized design space for scalable operator learning.
> > >
> > > Therefore, the three papers should not be viewed as near-duplicate variants of one method. Rather, they are complementary components of a unified program in which SPS studies the most physically constrained Hamiltonian setting through symplectic geometry, EPND studies geometry-driven reversible operator learning through Riemannian mechanics, and the Lie-algebraic Koopman paper studies the most practically flexible and computationally optimized regime through Lie-algebraic structure.
> > >
> > > We thank the reviewer again for the careful reading and thoughtful feedback, and note that, if the paper is accepted, we will make the code publicly available before the camera-ready deadline.

---

### Official Review · Reviewer_5Xrs · 2026-03-13

**Soundness:** 4
**Presentation:** 3
**Significance:** 4
**Originality:** 3
**Overall Recommendation:** 5
**Confidence:** 3

**Summary:**

The authors propose Euler–Poincaré Neural Dynamics (EPND), a dynamics-learning method in which state evolution is modeled through Lie-group motion rather than an unstructured neural update. States evolve via parallel transport along Lie-group geodesics governed by Euler–Arnold mechanics, introducing geometric biases toward symmetry, curvature, conservation, and reversibility. The method also exploits associativity in the group update to enable efficient parallel rollout. On molecular simulation tasks, the authors report improved accuracy, long-horizon stability, faster computation, and reconstruction of earlier states from terminal observations.

**Compliance With Llm Reviewing Policy:**

Affirmed.

**Key Questions For Authors:**

I do not have any key questions beyond the points I raised in strengths/weaknesses.

**Limitations:**

The authors do not note limitations. I suppose I would like to see a small section about when we could expect the assumptions of Lie-group motion to fail and if the model could be relaxed to explain those cases.

**Strengths And Weaknesses:**

Soundness: Serious mathematical grounding (although I’m not an expert in this area, as I have noted with my confidence score) and reasonably well motivated. The geometric setup is fairly clear and I found the associative scan idea clear and interesting. Benchmarking and ablations seem to follow best practice.

Presentation: Very good looking paper, but notation got hairy fast for a non expert. I would have benefited from a toy example in which the inductive biases of Lie-group motion were spelled out in something less complicated than molecular dynamics.

Significance: I think looking for good constraints for operator learning methods is a very important problem. The authors plumb some serious mathematical depths to find those–in this case–geometric constraints, which bear fruit in good performance. That performance in inverting long trajectories, the authors convincingly argue, comes from the preservation of geometric structure through parallel transport.

Originality: I am not an expert, but my understanding from studying this paper is that the synthesis of this Lie structure with flow learning is genuinely new.

---

> ### Author Rebuttal · Authors · 2026-03-30
>
> We are grateful for your encouraging evaluation of our work and for the recognition of its main strengths. We also appreciate the concrete questions and suggestions on the points that would benefit from clearer explanation. We have taken these points seriously and use this response to clarify the intended scope of the paper and the revisions we will make. We hope this response helps make the contribution and the planned improvements as clear as possible.
>
> **Q. I would have benefited from a toy example...**
>
> We thank you for this helpful suggestion. In the revised version, we will add a short toy example on $\mathfrak{gl}(2)$ before the molecular dynamics experiments. Concretely, we will consider a 2D state with $\xi_\theta(t) \in \mathfrak{gl}(2)$. Using two noncommuting generators makes the inductive bias explicit, since the model composes structured motions rather than arbitrary stepwise maps. We will use this toy example to first convey the basic noncommutative intuition and then briefly explain how the full method extends this idea through Lie-group exponentiation and the semidirect-product prefix scan.
>
> **Q. The authors do not note limitations…**
>
> We thank the reviewer for this insightful comment. Our method is primarily intended for settings in which the underlying dynamics can be naturally described by smooth Lie-group geodesic motion together with the associated parallel transport structure. In this regard, Proposition 4.2 and Appendix B characterize how far the proposed discrete scan-based construction may deviate from the exact transport, in terms of the step size and the regularity of the underlying geometry. In particular, Proposition 4.2 summarizes the cumulative mismatch over long rollouts, while Appendix B provides the supporting mathematical background and proof details.
>
> Accordingly, in regimes where the dynamics vary more rapidly or curvature effects are stronger, the mismatch between the discrete transport and the exact transport can accumulate more easily. To maintain the same level of geometric fidelity in such cases, a smaller step size may be required. More broadly, in settings where the geometric picture of smooth Lie-group motion itself becomes less appropriate, such as strongly dissipative dynamics or abrupt transitions, the suitability of the present formulation may be limited.
>
> Our current experiments are designed to demonstrate stability and do not yet include an empirical study that directly validates each term in Proposition 4.2. In the revised manuscript, we will make this point more explicit and add a short limitations discussion, together with future directions on more flexible geometric extensions and empirical evaluations that directly probe the bound.

---

> > ### Author Rebuttal · Reviewer_5Xrs · 2026-04-03
> >
> > I thank the authors for their reply and for their planned addition of a simple example and an updated limitations section. I think the toy example will be mostly useful if performance vs a model without the inductive bias is depicted. I will keep my score as it is reflecting my feeling that this paper should be accepted. However, I will reiterate that I am not an expert in this area and I can thus only make very general recommendations about this manuscript.

---

> > > ### Author Response · Authors · 2026-04-05
> > >
> > > We appreciate your positive assessment and your continued constructive engagement with the paper. We agree that the $\mathfrak{gl}(2)$ example becomes substantially more useful when it demonstrates the advantage of the Lie group inductive bias over an unstructured formulation. To maximize the clarity of this concept, we plan to consolidate several example-related suggestions from the reviewers into a single, comprehensive $\mathfrak{gl}$-centered presentation in the revised manuscript.

---

### Decision · Program_Chairs · 2026-04-30

**Decision:**

Accept (regular)

**Comment:**

This paper presents a novel geometric-mechanics framework for operator learning that combines Euler-Poincare/Lie-group dynamics with a parallel scan algorithm for scalable long-horizon simulation. From my reading of the reviews and the rebuttal comments/answers, reviewers tend to agree that the work is technically strong, original, and empirically sound, particularly (1) in its use of geometric structure as part of the model formulation, its (2) scan-based acceleration, and (3) reversible terminal value capability. While one reviewer remained hesitant, the concern centered primarily on exposition and accessibility rather than on soundness or invalidation of the technical claims.

From my perspective, the main weaknesses concern heavy notation, overly broad initial framing (e.g., physical consistency, interpretability etc.), insufficiently explicit discussion of limitations, and the need to better distinguish this work from related companion submissions! The author rebuttal addressed these issues credibly (for the most part), including commitments to narrow overstated claims, improve the concrete algorithmic exposition, add limitations and clarify experimental Lie algebra choices.

Overall, despite presentation weaknesses that should be addressed in the final version, the technical contribution appears substantive and sufficiently strong for acceptance. I recommend accept, contingent on the promised camera-ready revisions.